# MIMO-LP: A Multi-Input Multi-Output Framework for Subgraph-based Link Prediction

**Yixin Song**[1]  **Guangchi Liu**[1]  **Xiangyu Xu**[1]  **Shaofeng Li**[1]  **Zhen Ling**[1]  **Yiwei Wang**[2]  **Yujun Cai**[3]

## Abstract

Link prediction (LP) is a fundamental problem in graph learning and can be broadly categorized into node-based and subgraph-based approaches. Compared to node-based approaches, subgraph-based LP methods often achieve superior predictive performance by exploiting localized structural information, but suffer from significant efficiency bottlenecks due to the high computational cost of per-query subgraph message passing operations. To this end, we propose MIMO-LP, a *Multi-Input Multi-Output* (MIMO) framework that accelerates subgraph-based LP via *multiplexing*. Given a batch of query node pairs and their corresponding subgraphs extracted from a shared full graph, MIMO-LP *superposes* their message-passing processes into a shared latent space while ensuring minimal interference among them. This design enables MIMO-LP to multiplex multiple queries within a single forward pass during both training and inference, substantially reducing redundant message-passing computations in overlapping subgraph regions. Extensive experiments demonstrate that MIMO-LP achieves up to **44×** **speedup** over existing one-to-one subgraph-based methods, while maintaining comparable predictive performance.

## 1. Introduction

Graphs are ubiquitous data structures that model relationships between entities through nodes and edges. Given a graph and a query node pair, the link prediction (LP) task aims to infer the existence of a potential edge between them, with applications across a wide range of domains.

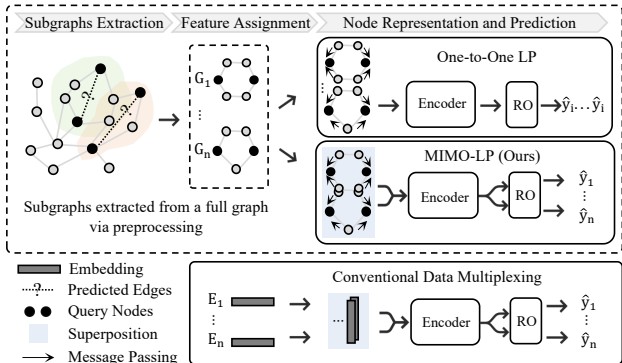

*Figure 1.* Illustration of the proposed **MIMO-LP** in comparison with conventional data multiplexing and one-to-one LP in the subgraph-based LP setting, where RO denotes a readout function. Notably, all operations introduced by MIMO-LP occur only at the node representation and prediction stage.

Link prediction (LP) methods can be broadly categorized into *node-based* and *subgraph-based* approaches. Node-based methods (Chen et al., 2020; Gasteiger, 2018; Yun et al., 2021) predict links by learning representations of query node pairs from information aggregated over the global graph. However, these methods often suffer from the *over-smoothing* problem (Topping et al., 2022), where excessive aggregation of global information leads to indistinguishable node representations, resulting in degraded predictive performance. In contrast, subgraph-based methods (Zhang & Chen, 2018; Tan et al., 2023; Du et al., 2025) leverage localized and contextual graph information (Mao et al., 2023), resulting in more accurate link predictions. However, such methods require extracting an individual subgraph and performing message-passing computations for each query, which incurs substantial computational overhead when the number of queries is large.

Recent advances in subgraph-based link prediction (LP) have explored various strategies to improve efficiency, primarily through distributed training (Besta & Hoefler, 2024; Kaler et al., 2023) and optimized subgraph extraction (Yin et al., 2022; 2023). In this work, we propose an orthogonal acceleration framework, MIMO-LP, a *Multi-Input Multi-Output* (MIMO) link prediction framework that multiplexes message-passing operations across multiple subgraphs dur-

[1]School of Computer Science and Engineering, Southeast University, Nanjing, China [2]Computer Science, University of California, Merced, USA [3]School of Electrical Engineering and Computer Science, The University of Queensland, Brisbane, Australia. Correspondence to: Guangchi Liu <loenixliu@gmail.com>.

*Proceedings of the 43rd International Conference on Machine Learning*, Seoul, South Korea. PMLR 306, 2026. Copyright 2026 by the author(s).

ing both training and inference. As illustrated in Fig. 1, given a batch of node-pair queries and their corresponding extracted subgraphs, MIMO-LP employs a multiplexing function to construct *superposed* representations of multiple message-passing signals, enabling them to propagate within a shared latent space with minimal interference. This design substantially reduces redundant message-passing computations in overlapping subgraph regions. Subsequently, MIMO-LP applies a de-multiplexing function to recover individualized representations for each node-pair query, which are then used for final link prediction. Different from the conventional subgraph-based LP paradigm, which processes each node-pair query and its associated subgraph in a one-to-one manner (see the upper part of Fig. 1), MIMO-LP executes message-passing operations for all queries within a batch simultaneously in a single forward pass, thereby substantially accelerating both training and inference.

To enable MIMO-LP, we address three technical challenges. The **first challenge** lies in how to multiplex the message-passing process to accelerate the subgraph-based LP models, where the principal computational bottleneck arises from inter-node message-passing rather than data propagation through neural network layers. To address this, MIMO-LP groups input subgraphs with overlapping edges from the full graph into batches, upon each of which a *union graph* is constructed, where redundant edges across subgraphs are deduplicated. On the union graph, message-passing signals from multiple subgraphs traversing shared regions are *superposed* into a single representation, with each signal preserved within a quasi-orthogonal subspace. This design enables message-passing operations for multiple subgraphs to be executed concurrently in a single pass over the union graph, thereby eliminating the redundant message-passing operations along the deduplicated edges.

The **second challenge** lies in how to *efficiently* group subgraphs into batches such that edge overlap among subgraphs is maximized, thereby enhancing acceleration by minimizing redundant message-passing computations within shared regions. Finding an exact solution to this objective is NP-complete, making it impractical for MIMO-LP, whose primary goal is to accelerate both training and inference. To overcome this, we demonstrate that maximizing edge overlap is approximately equivalent to minimizing the pairwise Jaccard distance between the edge sets of subgraphs. Since the Jaccard distance can be efficiently estimated using the MinHash algorithm applied to subgraph edge sets, MIMO-LP performs subgraph grouping through a clustering-based approach over the pairwise Jaccard distances, aiming to minimize intra-group Jaccard distance and thereby improve computational efficiency.

The **third challenge** lies in mitigating interference within the message-passing process as the number of multiplexed queries increases, which is critical for achieving higher acceleration. We identify that the key to interference mitigation is to (1) maintain orthogonality among individual signals and (2) prevent the phenomenon of information over-squashing (Topping et al., 2022), where excessive information from distinct nodes becomes superposed within the representation space of overlapping regions. To address the first issue, MIMO-LP introduces a generalizable modification that theoretically constrains the backbone graph encoder to behave as a linear transformation, thereby preserving the orthogonality of propagated signals. To address the second, MIMO-LP employs a random masking mechanism during both training and inference, wherein a subset of nodes in the union graph is randomly selected and their outgoing messages are zeroed out at each message-passing step. Together, these strategies enable MIMO-LP to scale to substantially larger multiplexing counts while maintaining competitive predictive accuracy.

We evaluate MIMO-LP on 14 link prediction datasets and 4 additional subgraph classification datasets spanning a range of scales, using four backbone subgraph-based LP models, and compare against three prior data multiplexing frameworks. Empirical results show that MIMO-LP delivers a **14–44× speedup** in training and inference over one-to-one subgraph-based LP baselines, while maintaining comparable predictive accuracy. Our contributions are summarized as follows:

- **New Research Perspective**: To our knowledge, MIMO-LP is the first to accelerate subgraph-based link prediction (LP) tasks via multiplexing techniques, bridging the gap between multiplex input data and message-passing operations.

- **Multiplexing toward Graph Learning**: We introduce techniques including union graph construction, overlap-aware subgraph batching, and random message dropout to multiplex the message-passing process, which is the dominant computational overhead in graph learning.

- **Model-Agnostic Design**: MIMO-LP is plug-and-play compatible with most subgraph-based LP models, providing a general framework across model architectures.

- **State-of-the-Art Performance**: MIMO-LP achieves **14–44× speedup** in both training and inference while maintaining near-identical predictive accuracy.

## 2. Related Works

### 2.1. Link Prediction Methods

Given a query node pair in a full graph, link prediction (LP) methods can be broadly categorized into node-based approaches (Chen et al., 2020; Song et al., 2023; Yun et al.,

2021) and subgraph-based approaches (Zhang & Chen, 2018; Teru et al., 2020; Tan et al., 2023; Du et al., 2025). Node-based methods apply a graph encoder to learn representations of query nodes from the full graph and leverage these representations for link prediction. However, they often suffer from limited accuracy, as they aggregate excessive information from the full graph, leading to the *over-squashing* problem (Topping et al., 2022). To address this limitation, subgraph-based methods (Zhang & Chen, 2018; Teru et al., 2020; Tan et al., 2023; Du et al., 2025; Shomer et al., 2024) perform link prediction based on localized subgraph structures induced around each query node pair. While such localized modeling improves predictive performance, it introduces substantial computational overhead, since each query incurs separate subgraph extraction and message-passing computations. Recent studies have proposed several strategies to accelerate subgraph-based LP. For example, distributed training frameworks parallelize GNN encoders across multiple servers (Besta & Hoefler, 2024; Kaler et al., 2023), while Yin *et al.* (Yin et al., 2022) employ random-walk-based positional encodings to improve feature contextualization efficiency. More recently, NCN (Wang et al., 2024) accelerates training via an efficient readout mechanism that enhances pairwise information aggregation. However, none of these methods achieve acceleration from a *Multi-Input Multi-Output* (MIMO) perspective—an orthogonal direction that motivates the design of MIMO-LP.

### 2.2. Data Multiplexing

Inspired by multiplexing techniques widely used in communication systems, recent advances in machine learning (Havasi et al., 2021; Alexandre Ramé & Cord, 2021; Elhage et al., 2022) have shown that neural networks can process multiple inputs simultaneously within a single forward pass by encoding different input signals into orthogonal subspaces of a shared representation, a mechanism referred to as *superposition*. Building on this idea, Murahari *et al.* (Murahari et al., 2022), Menet *et al.* (Menet et al., 2024), and Xu *et al.* (Xu et al., 2024) introduce data multiplexing techniques for Transformers and large language models (LLMs). In these approaches, multiple input embeddings are multiplexed into a single superposed representation, enabling the backbone network to process several inputs in one forward pass and thereby significantly accelerating both training and inference. However, as illustrated in Fig. 1, these methods are not directly applicable to graph learning tasks, where message-passing operations dominate the computational cost. This limitation arises because existing approaches primarily focus on multiplexing input representations through neural network layers, rather than the underlying message-passing process. In contrast, MIMO-LP is the first framework to accelerate subgraph-based LP by multiplexing message-passing operations themselves.

## 3. Preliminaries

In this section, we first revisit the standard one-to-one pipeline for subgraph-based link prediction. Building on this foundation, we then introduce the tasks that MIMO-LP is designed to address.

### 3.1. Subgraph-based Link Prediction Pipeline

Let $G = (\mathcal{V}, \mathcal{E})$ be an undirected graph, where $\mathcal{V}$ denotes the set of nodes and $\mathcal{E}$ the set of observed edges. Given a set of node-pair queries $\mathcal{T} = \{(u_i, v_i) \mid u_i, v_i \in \mathcal{V}\}$, where $i$ indexes the query pairs, a standard one-to-one subgraph-based link prediction pipeline can be decomposed into three steps. The first step is *subgraph extraction*, i.e., extracting an induced subgraph $G_i$ from $G$ based on the query node pair $(u_i, v_i)$ and their neighborhoods:

$$G_i = \mathcal{F}_{\text{extr}}(u_i, v_i, G),$$

where $G_i$ is the $i$-th induced subgraph of $G$ associated with the query pair $(u_i, v_i)$. The second step is *feature assignment*, i.e., assigning a contextualized feature vectors to nodes in each subgraph $G_i$:

$$\mathbf{X}_{G_i} = \mathcal{F}_{\text{contx}}(G_i), \tag{1}$$

where $\mathbf{X}_{G_i} = [\mathbf{x}_1, \mathbf{x}_2, \ldots, \mathbf{x}_{|\mathcal{V}_{G_i}|}] \in \mathbb{R}^{|\mathcal{V}_{G_i}| \times d}$ is a matrix storing the contextual node-feature vectors for subgraph $G_i$. Notably, a node $v \in \mathcal{V}$ may have different contextual features across subgraphs $G_i$ and $G_j$ if it appears in both. The third step is *node representation and prediction*, i.e., applying an $L$-layer ($L > 0$) stacked encoder $\mathcal{F}_{\text{enc}}$ on the subgraph with contextualized node features to obtain node representations. The $l$-th layer representation can be obtained by

$$\mathbf{H}_{G_i}^{(l)} = \begin{cases} \mathcal{F}_{\text{enc}}^{(l)}(\mathbf{X}_{G_i}, G_i) & \text{if} \quad l = 1 \\ \mathcal{F}_{\text{enc}}^{(l)}(\mathbf{H}_{G_i}^{(l-1)}, G_i) & \text{if} \quad 1 < l \leq L \end{cases} \tag{2}$$

where $\mathbf{H}_{G_i}^{(l)} = [\mathbf{h}_1, \mathbf{h}_2, \ldots, \mathbf{h}_{|\mathcal{V}_{G_i}|}] \in \mathbb{R}^{|\mathcal{V}_{G_i}| \times d}$ represents the hidden node representation learned during the message-passing process. Finally, predictions for the query pairs $(u_i, v_i) \in \mathcal{T}$ are obtained by pooling the output of the last layer:

$$\hat{y}_i = \text{Readout}(\mathbf{H}_{G_i}^{(L)}),$$

where Readout denotes a graph-level pooling function.

Notably, MIMO-LP follows the standard subgraph-based link prediction pipeline for subgraph extraction and feature assignment, and achieves MIMO-based acceleration at the third step, i.e., node representation and prediction.

### 3.2. Task Definition

The key to accelerating the subgraph-based link prediction (LP) pipeline lies in reducing the message-passing operations during pre-query subgraph processing, which account

for the majority of the computational overhead in subgraph-based methods. To maximize acceleration while maintaining predictive accuracy by multiplexing message-passing operations across different subgraphs, the design of MIMO-LP is decomposed into two fundamental tasks.

**Task I: Subgraph Batching.** The first task is to efficiently group subgraphs into batches such that the overlap of edges across subgraphs is maximized. The purpose is to deduplicate message-passing operations in overlapping regions as much as possible, thereby achieving maximal computational acceleration.

Formally, given a set of $N$ subgraphs $\mathcal{G} = \{G_i \mid 1 \leq i \leq N\}$ extracted by node-pair queries $\mathcal{T} = \{(u_i, v_i) \mid u_i, v_i \in \mathcal{V}, 1 \leq i \leq N\}$ from a full graph $G$, we partition the subgraphs into $K$ disjoint batches $\mathcal{B} = \{\mathcal{B}_k \mid 1 \leq k \leq K\}$, where $\mathcal{B}_k \subset \mathcal{G}$ denotes the $k$-th batch. For each batch $\mathcal{B}_k$, we construct a *union graph* $G_{\mathcal{B}_k}$ by merging all subgraphs in the batch:

$$G_{\mathcal{B}_k} = G\Big[\bigcup_{G_i \in \mathcal{B}_k} \mathcal{V}_{G_i}\Big], \tag{3}$$

where $\mathcal{V}_{G_i}$ denotes the node set of subgraph $G_i$. By performing message passing over the union graph rather than over individual subgraphs, redundant computations on overlapping edges can be effectively eliminated. We quantify this elimination using the *Edge Reduction Ratio* (ERR), defined as the ratio of eliminated edges to the total number of edges across all subgraphs:

$$\text{ERR} = \frac{\sum_{\mathcal{B}_k \in \mathcal{B}} \left(\sum_{G_i \in \mathcal{B}_k} |\mathcal{E}_{G_i}| - |\mathcal{E}_{G_{\mathcal{B}_k}}|\right)}{\sum_{G_i \in \mathcal{G}} |\mathcal{E}_{G_i}|}, \tag{4}$$

where $\mathcal{E}_{G_i}$ and $\mathcal{E}_{G_{\mathcal{B}_k}}$ denote the edge sets of $G_i$ and $G_{\mathcal{B}_k}$, respectively. The term $\sum_{G_i \in \mathcal{B}_k} |\mathcal{E}_{G_i}| - |\mathcal{E}_{G_{\mathcal{B}_k}}|$ quantifies the total number of overlapping edges eliminated within the $k$-th batch. Under this setting, **Task I** is formulated as the following optimization problem, which seeks an optimal batch assignment $\mathcal{B}^\star$ that maximizes ERR:

$$\begin{aligned} &\text{Maximize:} \quad \text{ERR}, \\ &\text{Subject to:} \quad |\mathcal{V}_{G_{\mathcal{B}_k}}| \leq \text{B}, \bigcup_{\mathcal{B}_k \subset \mathcal{G}} \mathcal{B}_k = \mathcal{G}, \bigcap_{\mathcal{B}_k \subset \mathcal{G}} \mathcal{B}_k = \emptyset, \end{aligned} \tag{5}$$

where B denotes the maximum allowable number of nodes in each union graph.

**Task II: Multiplexing and De-multiplexing Functions.** With the optimal subgraph batching (denoted as $\mathcal{B}$ for simplicity) obtained in Task I, **Task II** is to design a *multiplexing* and a corresponding *de-multiplexing* function upon the standard one-to-one LP pipeline (see Eq. 2). These functions enable simultaneous execution of message-passing for all subgraphs within a batch by leveraging their union graph

and a shared backbone encoder as

$$\hat{y}_1, \ldots, \hat{y}_{|\mathcal{B}_k|} = \text{Readout}\Big(\mathcal{F}_{\text{DM}}\Big(\mathbf{H}^{(L)}_{G_{\mathcal{B}_k}}\Big)\Big),$$

where:

$$\mathbf{H}^{(l)}_{G_{\mathcal{B}_k}} = \begin{cases} \mathcal{F}^{(1)}_{\text{enc}}\Big(\mathcal{F}_{\text{M}}\Big(\mathbf{X}_{G_1}, \ldots, \mathbf{X}_{G_{|\mathcal{B}_k|}}\Big), G_{\mathcal{B}_k}\Big), & l = 1, \\ \mathcal{F}^{(l)}_{\text{enc}}\Big(\mathbf{H}^{(l-1)}_{G_{\mathcal{B}_k}}, G_{\mathcal{B}_k}\Big), & l > 1. \end{cases} \tag{6}$$

In Eq. 6, $\mathcal{F}_{\text{M}} : \mathbb{R}^{\sum_{G_i \in \mathcal{B}_k} |\mathcal{V}_{G_i}| \times d} \to \mathbb{R}^{|\mathcal{V}_{G_{\mathcal{B}_k}}| \times d}$ denotes the multiplexing function that superposes the message-passing signals from multiple subgraphs into a shared latent space, whose dimension is consistent with that of the original graph encoder $\mathcal{F}_{\text{enc}}$. Conversely, $\mathcal{F}_{\text{DM}} : \mathbb{R}^{|\mathcal{V}_{G_{\mathcal{B}_k}}| \times d} \to \mathbb{R}^{\sum_{G_i \in \mathcal{B}_k} |\mathcal{V}_{G_i}| \times d}$ denotes the de-multiplexing function, which projects the shared representations back into their respective subgraph-specific spaces. $1 \leq l \leq L$ denotes the layer of the graph model. Finally, $\hat{y}_1, \ldots, \hat{y}_n$ denote the predictions obtained by applying a readout (pooling) function to the de-multiplexed representations. These predictions are required to align with the ground-truth labels. Formally, this objective can be expressed as:

$$\text{Minimize:} \quad \sum_{1 \leq i \leq N} \text{loss}(y_i, \hat{y}_i), \tag{7}$$

where $\text{loss}(\cdot, \cdot)$ denotes the LP loss function.

## 4. Methodology

Building upon the two tasks introduced in Section 3.2, we now present the overall design of MIMO-LP, as illustrated in Fig. 2. Specifically, in **Section 4.1**, we introduce a clustering-based batching method that efficiently groups subgraphs into approximately optimal batches, thereby addressing **Task I**. To address **Task II**, we develop a multiplexing function and a de-multiplexing function, which are described in **Section 4.2** and **Section 4.4**, respectively. In addition, to mitigate potential interference among different message-passing processes, we introduce two types of strategies that enhance the robustness of MIMO-LP. These strategies are detailed in **Section 4.3**. A theoretical justification of the multiplexing and interference-mitigation mechanisms is provided in Appendix B.1.

### 4.1. Subgraph Batching

Since finding the exact solution to the optimal subgraph batching problem in Eq. 5 is NP-complete (bin-packing problem), we adopt a clustering-based approximation to obtain an efficient solution.

Specifically, the clustering method aims to group the subgraph set $\mathcal{G}$ such that the Jaccard Index between the edge sets

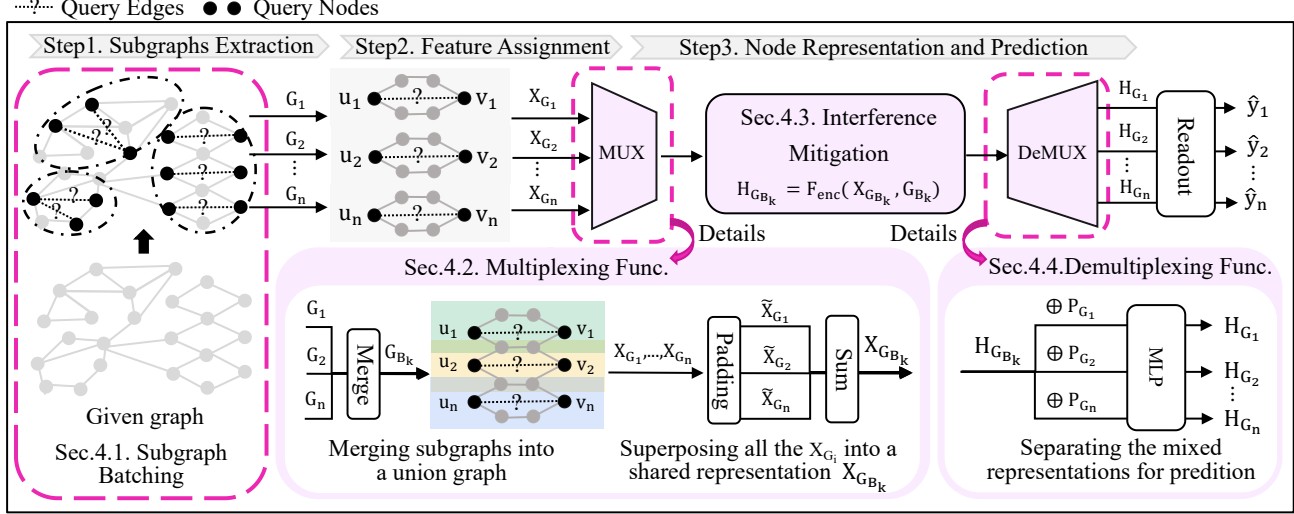

*Figure 2.* The overview of the proposed MIMO-LP. The clustering-based batching maximizes ERR among a batch of subgraphs (details in Sec.4.1). The multiplexing layer employs Merge to construct the union graph $G_{\mathcal{B}_k}$ and employs Padding to standardize the shapes of the $n$ node structural embedding matrices $\mathbf{X}_{G_i}$ as $\widetilde{\mathbf{X}}_{G_i}$, finally employs Sum to superimpose these feature matrices as the multiplexing input $\mathbf{X}_{G_{\mathcal{B}_k}}$ (details in Sec.4.2). The graph encoder encodes the multiplexing representation to a hidden representation $\mathbf{H}_{G_{\mathcal{B}_k}}$ (details in Sec.4.3). The demultiplexing layer employs a shared MLP to recover individual subgraph representations for prediction (details in Sec.4.4).

of intra-cluster subgraphs, i.e., $\mathcal{J}(\mathcal{E}_{G_m}, \mathcal{E}_{G_n})$, $\forall G_m, G_n \in \mathcal{B}_k$, is greater than that between inter-cluster subgraphs. A theoretical justification demonstrating that finding the optimal $\mathcal{B}^\star$ maximizing ERR is approximately equivalent to clustering with respect to the Jaccard Index is provided in Appendix A.1. The Jaccard Index can be computed efficiently using the MinHash method (Wu et al., 2022):

$$
\begin{aligned}
\mathcal{J}(\mathcal{E}_{G_m}, \mathcal{E}_{G_n}) &= \frac{|\mathcal{E}_{G_m} \cap \mathcal{E}_{G_n}|}{|\mathcal{E}_{G_m} \cup \mathcal{E}_{G_n}|} \\
&= \Pr\Big(\min\big(\pi^{(t)}(\mathcal{E}_{G_m})\big) = \min\big(\pi^{(t)}(\mathcal{E}_{G_n})\big)\Big),
\end{aligned}
\tag{8}
$$

where $\pi^{(t)}(\mathcal{E}_{G_m})$ denotes the set obtained by applying $t$ randomized hash functions to $\mathcal{E}_{G_m}$, and $\min(\pi^{(t)}(\mathcal{E}_{G_m}))$ denotes the minimum hash value in the set. Here, $\Pr(\cdot)$ represents the probability of two sets producing the same MinHash value (i.e., a hash collision). In practice, MIMO-LP employs the $k$-means algorithm with the Jaccard Index as the similarity metric and $K$ as the number of clusters. Consequently, MIMO-LP partitions $\mathcal{G}$ into $K$ subgraph batches, denoted as $\mathcal{B}^\star = \{\mathcal{B}_j^\star \mid 1 \le j \le K\}$, together with their associated query node pairs $\mathcal{T}^\star = \{\mathcal{T}_j^\star \mid 1 \le j \le K\}$. For each batch $\mathcal{B}_j^\star \in \mathcal{B}^\star$, a union graph $G_{\mathcal{B}_j^\star}$ can be constructed, enabling substantial reduction in redundant message-passing operations. For simplicity, we use $\mathcal{B}$ to denote the clustering result throughout the rest of this paper.

Notably, both clustering and union graph construction process incur negligible overhead compared to the overall pipeline, as confirmed by the detailed complexity analysis (Appendix D.1) and runtime breakdown (Appendix E.5).

The cluster number $K$ is determined by the desired multiplexing query count $n$ and the total number of node-pair queries, i.e., $K = \frac{1}{n}|\mathcal{T}|$. Intuitively, increasing $n$ reduces the value of $K$, thereby increasing efficiency but at the risk of degraded accuracy due to greater interference among queries. In Appendix A.2, we present an efficient algorithm for determining the value of $n$.

## 4.2. Multiplexing Function

The multiplexing function $\mathcal{F}_{\mathrm{M}}$ superposes contextualized feature vectors of overlapping nodes into a shared representation whose dimension is consistent with that of the backbone graph encoder, enabling message-passing signals from multiple node-pair queries to propagate simultaneously within the constructed union graph.

Following the second step of the subgraph-based LP pipeline (Eq. 1), for a batch of subgraphs $\mathcal{B}_k = \{G_i \mid 1 \le i \le n\}$, we obtain $n$ sets of node representations, where each set $\mathbf{X}_{G_i}$ stores the contextualized feature vectors of all nodes in subgraph $G_i$. To facilitate superposition in the union graph, we first construct a contextualized feature matrix $\widetilde{\mathbf{X}}_{G_i} \in \mathbb{R}^{|\mathcal{V}_{G_{\mathcal{B}_k}}| \times d}$ for each subgraph:

$$
\widetilde{\mathbf{X}}_{G_i}[\mathbf{m}_{G_i}] = \mathbf{X}_{G_i},
\tag{9}
$$

where $\mathbf{m}_{G_i}$ indexes the entries of the union graph nodes that appear in subgraph $G_i$. For nodes not present in $G_i$, their entries are assigned zero, i.e., $\widetilde{\mathbf{X}}_{G_i}[\overline{\mathbf{m}}_{G_i}] = \mathbf{0}$.

Given the contextualized feature matrices of all subgraphs

in a batch $\{\widetilde{\mathbf{X}}_{G_i} \mid G_i \in \mathcal{B}_k\}$, the superposition of subgraph-specific node features within the union graph is expressed as a weighted summation:

$$\mathbf{X}_{G_{\mathcal{B}_k}} = \mathbf{D}^{-1} \sum_{G_i \in \mathcal{B}_k} \left( \mathbf{\Phi}_{G_i} \odot \widetilde{\mathbf{X}}_{G_i} \right), \qquad (10)$$

where $\mathbf{\Phi}_{G_i} \in \mathbb{R}^{|\mathcal{V}_{G_{\mathcal{B}_k}}| \times d}$ is a Gaussian noise matrix, $\odot$ denotes the Hadamard product, and $\mathbf{D}^{-1} = \mathrm{diag}\left( \left( \sum_{G_i \in \mathcal{B}_k} \mathbf{m}_{G_i} \right)^{-1} \right)$ is a diagonal normalization matrix whose entries normalize the superposed vectors for each node in the union graph. In Eq. 10, the term $\mathbf{\Phi}_{G_i} \odot \widetilde{\mathbf{X}}_{G_i}$ projects the representation $\widetilde{\mathbf{X}}_{G_i}$ into a new subspace determined by the noise matrix $\mathbf{\Phi}_{G_i}$. Prior work (Menet et al., 2024; Alexandre Ramé & Cord, 2021) demonstrates that if two Gaussian noise matrices $\mathbf{\Phi}_{G_i}$ and $\mathbf{\Phi}_{G_j}$ are quasi-orthogonal, then their corresponding projected vectors, $\mathbf{\Phi}_{G_i} \odot \widetilde{\mathbf{X}}_{G_i}$ and $\mathbf{\Phi}_{G_j} \odot \widetilde{\mathbf{X}}_{G_j}$, are also quasi-orthogonal (a theoretical justification is provided in Assumption B.1 of Appendix B.1). This property ensures that the contextualized feature vectors of the same node across different subgraphs can be superposed into a single representation without destructive interference, while preserving the dimensionality of the feature vectors.

## 4.3. Interference Mitigation

Despite of the orthogonality of superposed input features after applying the multiplexing function, the interference, which can lead to a degradation in prediction accuracy, could still happen. In this section, we introduce two types of strategies mitigating this issue.

The first type of strategies focuses on preserving the orthogonality of signals across different node-pair queries. While multiplexing achieves orthogonality for the input features, this property can be compromised by the non-linear transformations within the backbone graph encoder. We address this by introducing a plug-in yet generalizable modification that enforces the encoder $\mathcal{F}_{\mathrm{enc}}$ to be approximately linear, with the modified encoder denoted as $\mathcal{F}_{\mathrm{l\text{-}enc}}$. Intuitively, since Eq. 10 ensures the quasi-orthogonality of the inputs to $\mathcal{F}_{\mathrm{l\text{-}enc}}$, this property will be preserved at the output as long as $\mathcal{F}_{\mathrm{l\text{-}enc}}$ remains a linear transformation (Havasi et al., 2021). Specifically, MIMO-LP employs the Parametric Rectified Linear Unit (He et al., 2015) (PReLU) as a semi-linear activation function in the backbone encoder, replacing more commonly used non-linear activations such as ReLU:

$$\mathrm{PReLU}(x) = \max(x, 0) + b \min(x, 0) \qquad (11)$$

where the trainable parameter $b \in [-1, 1]$ controls the degree of linearity, and $b = 1$ indicates fully linear behavior. A theoretical justification of why this strategy preserves approximate linearity across most graph encoder architectures is provided in Appendix B.1. In addition, MIMO-LP

applies an orthogonality regularization term to the weight matrix of each encoder layer to encourage orthogonal transformations (Bansal et al., 2018; Huang et al., 2017):

$$\mathcal{L}_{\mathrm{O}} = \left\| \mathbf{W}^{\top} \mathbf{W} - \mathbf{I} \right\|_F^2, \qquad (12)$$

where $\mathbf{W}$ denotes the layer weight matrix and $\mathbf{I}$ is the identity matrix.

The second type of strategies acknowledges that absolute orthogonality is difficult to guarantee, and thus interference may still occur. We attribute this interference to the over-squashing problem (Topping et al., 2022; Di Giovanni et al., 2023) in graph neural networks, where excessive and heterogeneous information from distinct nodes becomes compressed into the representation space of overlapping nodes, leading to a degradation in predictive accuracy. To alleviate this, MIMO-LP introduces a random masking mechanism that selectively zeros out messages from certain nodes in the union graph. Specifically, before aggregation at each graph encoder layer, a mask $\mathbf{M}_\rho \in \mathbb{R}^{|\mathcal{V}_{G_{\mathcal{B}_k}}| \times d}$ is applied to the node features:

$$\mathbf{H}_{G_{\mathcal{B}_k}} = \mathcal{F}_{\mathrm{l\text{-}enc}} \left( \mathbf{M}_\rho \odot \mathbf{X}_{G_{\mathcal{B}_k}}, G_{\mathcal{B}_k} \right), \qquad (13)$$

where $\mathcal{F}_{\mathrm{l\text{-}enc}}$ is an arbitrary graph encoder, each row of $\mathbf{M}_\rho[i, :] = \mathbf{0}$ with probability $\rho$, and otherwise remains unchanged. This random masking is applied during both training and inference. In our experiments (see Appendix E.3 for details), we empirically set the hyperparameter $\rho$ to $0.2$ and found this value to be consistent across different datasets.

## 4.4. De-multiplexing Function

Given the superposed representation at the output layer, $\mathbf{H}_{G_{\mathcal{B}_k}}$, MIMO-LP applies a de-multiplexing function $\mathcal{F}_{\mathrm{DM}}$ to disentangle the individual signals and generate predictions for each query.

Motivated by the concept of aggregated learning (Soflaei et al., 2023), which demonstrates that fine-grained information can be effectively recovered from superposed representations through task-specific optimization (a formal justification is provided in Assumption B.2 of Appendix B.1), we introduce an index matrix $\mathbf{P}_{G_i} \in \mathbb{R}^{|\mathcal{V}_{G_{\mathcal{B}_k}}| \times d_p}$ for each $G_i$. The matrix is concatenated with $\mathbf{H}_{G_{\mathcal{B}_k}}$ and fed into a shared MLP layer to yield the individual representation $\widetilde{\mathbf{H}}_{G_i} \in \mathbb{R}^{|\mathcal{V}_{G_{\mathcal{B}_k}}| \times d}$:

$$\widetilde{\mathbf{H}}_{G_i} = \mathrm{MLP}\left( \mathbf{P}_{G_i} \oplus \mathbf{H}_{G_{\mathcal{B}_k}} \right), \qquad (14)$$

where $\mathbf{P}_{G_i}$ is trainable with respect to the link prediction task loss. Intuitively, $\mathbf{P}_{G_i}$ serves as an additional prior associated with the query, enabling more distinct representation separation for the corresponding link predictions. A visualization of the separated representations is provided in

Appendix C. Finally, each $\mathbf{H}_{G_i}$ is extracted from $\widetilde{\mathbf{H}}_{G_i}$ via indexing with $\mathbf{m}_{G_i}$, and each $\mathbf{H}_{G_i}$ is then processed by a readout function to generate the final prediction.

A theoretical justification for the multiplexing and de-multiplexing design, grounded in the functional additivity of the modified message-passing operations, is provided in Proposition B.3 in Appendix B.1.

### 4.5. Training and Inference

With the readout operation in Eq. 14, we obtain predictions $\{\hat{y}_i \mid 1 \leq i \leq n\}$ for the $n$ queries within a batch. Taking the corresponding edge labels $\{y_i \mid 1 \leq i \leq n\}$ as ground truth, the pipeline defined in Eq. 6 is trained with the following loss function:

$$\mathcal{L}_{\text{Total}} = \mathcal{L}_{\text{CE}} + \mathcal{L}_O,$$
$$\text{where:} \quad \mathcal{L}_{\text{CE}} = -\frac{1}{n} \sum_{i=1}^{n} \sum_{c=1}^{C} y_{i,c} \log(\hat{y}_{i,c}). \tag{15}$$

Here $\mathcal{L}_{\text{CE}}$ denotes the cross-entropy loss between predictions and labels and $\mathcal{L}_O$ is the orthogonality regularization loss introduced in Eq. 12. The learnable components include the graph encoder $\mathcal{F}_{\text{l-enc}}$ defined in Eq. 13, the index matrix $\mathbf{P}$, and the MLP used in Eq. 14.

During inference, MIMO-LP first extracts subgraphs corresponding to all node-pair queries and clustering them into batches using the method introduced in Section 4.1. Each batch is fed into the trained pipeline defined in Eq. 6, which performs a single forward pass to simultaneously compute a batch of predictions $\hat{y}_1, \ldots, \hat{y}_n$.

A detailed complexity analysis (Appendices D.1 and D.2) and empirical evaluation (Section 5) confirm that, although MIMO-LP introduces only trivial overhead, it eliminates a substantial amount of redundant computation on overlapping edges (exceeding 70% as measured by ERR in most real-world datasets) during the node representation and prediction stages of one-to-one subgraph-based LP pipelines, thereby achieving significant acceleration in both training and inference.

## 5. Evaluation

In this section, we evaluate the performance of **MIMO-LP** and conduct ablation studies on the proposed mechanisms. Our code is available at: `https://github.com/aike333/MIMO-LP-/tree/main`.

### 5.1. Experimental Setup

**Environment.** All experiments are conducted on a cloud server equipped with 12 vCPUs (Intel(R) Xeon(R) Platinum 8255C CPU @ 2.50GHz), one NVIDIA RTX 2080 Ti GPU (11GB), using Python 3.8 and PyTorch 2.1.0.

**Compared Baselines.** We evaluate MIMO-LP by integrating it into five representative subgraph-based backbone models: **SEAL** (Zhang & Chen, 2018), **PS2** (Tan et al., 2023), **SMA** (Zhao et al., 2025), **NCN** (Wang et al., 2024), and **LPFormer** (Shomer et al., 2024). The resulting variants are denoted as **M-SEAL**, **M-PS2**, **M-SMA**, **M-NCN**, and **M-LPFormer**, respectively. We compare these models against three existing data multiplexing frameworks: Data-Mux(DM) (Murahari et al., 2022), MIMONets(KM) (Menet et al., 2024), and RevMux(RM) (Xu et al., 2024), which results in 15 baselines denoted in similar style, as shown in Table 1. Notably, since DM, KM, and RM are designed to multiplex input vectors rather than message-passing processes, we use the contextualized feature vectors, i.e., $\mathbf{X}_{G_i}$, of multiple subgraphs as the input for these methods. For fair comparison, all backbone encoders adopt a unified three-layer architecture.

**Datasets.** We conduct experiments on 14 benchmark link prediction datasets of varying scales, with node counts ranging from 1.5k to 91M, where detailed descriptions are provided in Appendix E. Due to space limitations, we present results on 5 representative datasets in this section, including **Yeast** (Von Mering et al., 2002), **DrugBank** (Newman, 2006), **Friendster** (Yang & Leskovec, 2012), **Collab** (Hu et al., 2020), and **PPA** (Hu et al., 2020). Similar results on the remaining datasets are reported in Appendix E.4. To further evaluate the generalizability of MIMO-LP to other tasks, we additionally report similar results on 4 subgraph classification benchmarks, as detailed in Appendix E.6. For all experiments, the train–test split ratio is set to 8:2.

**Evaluation Metrics.** We use three evaluation metrics. The first is **Speedup Ratio (SR)**, which measures acceleration in runtime and is defined as $\text{SR} = \frac{\text{Runtime of } X}{\text{Runtime of M-}X}$, where $X$ is the original model and M-$X$ is its multiplexing variant. The runtime includes both training and inference time, as well as the subgraph batching and union graph construction time for MIMO-LP. The second is **AUC/HR@50/HR@100**, which is used to evaluate link prediction performance. For Yeast, DrugBank and Friendster, we follow Zhang et al. (Zhang & Chen, 2018) and use AUC. For Collab and PPA, we follow Shomer et al. (Shomer et al., 2024) and use HR@50 and HR@100, respectively. The last one is **Edge Reduction Ratio (ERR)**, which measures edge (message-passing operation) reduction ratio and is defined in Eq. 5. To reduce randomness, the results are averaged by running independently 10 times.

### 5.2. Overall Performance

We first present the overall performance of MIMO-LP compared with the baselines, as summarized in Table 1. In the experiments, we fix the multiplexing count (n) to compare

*Table 1.* Model comparison using SEAL, PS2, SMA, NCN and LPFormer (LPFm) as backbone model. n indicates the number of merged subgraphs. Speedup Ratio (SR) are reported against the n = 1 setting. Best results in bold.

| Models | Datasets | | | | | | | | | | | | | | | | | | | |
| --- | --- | --- | --- | --- | --- | --- | --- | --- | --- | --- | --- | --- | --- | --- | --- | --- | --- | --- | --- | --- |
| | Yeast | | | | DrugBank | | | | Friendster | | | | Collab | | | | PPA | | | |
| | n | AUC | SR↑ | ERR | n | AUC | SR↑ | ERR | n | AUC | SR↑ | ERR | n | HR@50 | SR↑ | ERR | n | HR@100 | SR↑ | ERR |
| **M-SEAL** (Ours) | 50 | **94.87**$_{\pm0.50}$ | 16 | 0.78 | 50 | **96.79**$_{\pm0.80}$ | 25 | 0.80 | 50 | 90.38$_{\pm0.50}$ | 15 | 0.74 | 100 | 64.29$_{\pm0.60}$ | 34 | 0.81 | 100 | 48.80$_{\pm0.50}$ | 44 | 0.86 |
| DM-SEAL | 2 | 86.68$_{\pm1.00}$ | 2 | - | 2 | 80.01$_{\pm0.90}$ | 2 | - | 2 | 81.38$_{\pm2.00}$ | 2 | — | 2 | 61.30$_{\pm1.50}$ | 2 | - | 2 | 43.28$_{\pm1.50}$ | 2 | - |
| KM-SEAL | 2 | 85.30$_{\pm1.00}$ | 2 | - | 2 | 80.58$_{\pm1.20}$ | 2 | - | 2 | 80.30$_{\pm2.00}$ | 2 | - | 2 | 61.24$_{\pm1.50}$ | 2 | - | 2 | 42.16$_{\pm1.50}$ | 2 | - |
| RM-SEAL | 2 | 89.21$_{\pm1.00}$ | 2 | - | 2 | 80.39$_{\pm1.00}$ | 2 | - | 2 | 83.27$_{\pm2.00}$ | 2 | - | 2 | 61.26$_{\pm1.50}$ | 2 | - | 2 | 41.64$_{\pm1.50}$ | 2 | - |
| SEAL | 1 | 95.46$_{\pm0.20}$ | 1 | 0 | 1 | 96.84$_{\pm0.10}$ | 1 | 0 | 1 | 92.71$_{\pm0.30}$ | 1 | 0 | 1 | 64.38$_{\pm0.40}$ | 1 | 0 | 1 | 48.83$_{\pm0.30}$ | 1 | 0 |
| **M-PS2** (Ours) | 50 | 95.29$_{\pm0.60}$ | 16 | 0.78 | 50 | **97.25**$_{\pm0.80}$ | 25 | 0.80 | 50 | 91.87$_{\pm0.50}$ | 15 | 0.74 | 100 | 65.43$_{\pm0.60}$ | 34 | 0.81 | 100 | 49.16$_{\pm0.50}$ | 44 | 0.86 |
| DM-PS2 | 2 | 86.40$_{\pm1.50}$ | 2 | - | 2 | 81.30$_{\pm1.20}$ | 2 | - | 2 | 81.31$_{\pm2.00}$ | 2 | - | 2 | 61.75$_{\pm2.00}$ | 2 | - | 2 | 41.37$_{\pm2.00}$ | 2 | - |
| KM-PS2 | 2 | 86.24$_{\pm1.50}$ | 2 | - | 2 | 81.37$_{\pm1.20}$ | 2 | - | 2 | 80.26$_{\pm2.00}$ | 2 | - | 2 | 61.02$_{\pm2.00}$ | 2 | - | 2 | 43.27$_{\pm1.50}$ | 2 | - |
| RM-PS2 | 2 | 89.40$_{\pm1.50}$ | 2 | - | 2 | 85.10$_{\pm1.20}$ | 2 | - | 2 | 83.54$_{\pm2.00}$ | 2 | - | 2 | 61.79$_{\pm2.00}$ | 2 | - | 2 | 43.56$_{\pm2.00}$ | 2 | - |
| PS2 | 1 | 96.07$_{\pm0.18}$ | 1 | 0 | 1 | 97.31$_{\pm0.30}$ | 1 | 0 | 1 | 93.52$_{\pm0.30}$ | 1 | 0 | 1 | 65.48$_{\pm0.40}$ | 1 | 0 | 1 | 49.17$_{\pm0.30}$ | 1 | 0 |
| **M-SMA** (Ours) | 50 | 95.64$_{\pm0.60}$ | 16 | 0.78 | 50 | 96.80$_{\pm0.30}$ | 23 | 0.80 | 50 | 91.33$_{\pm0.90}$ | 15 | 0.74 | 100 | 64.31$_{\pm0.70}$ | 33 | 0.81 | 100 | 50.29$_{\pm0.60}$ | 42 | 0.86 |
| DM-SMA | 2 | 85.79$_{\pm1.50}$ | 2 | - | 2 | 80.74$_{\pm1.00}$ | 2 | - | 2 | 80.14$_{\pm2.00}$ | 2 | - | 2 | 61.67$_{\pm2.00}$ | 2 | - | 2 | 45.35$_{\pm2.00}$ | 2 | - |
| KM-SMA | 2 | 85.41$_{\pm1.50}$ | 2 | - | 2 | 80.57$_{\pm1.80}$ | 2 | - | 2 | 80.71$_{\pm2.00}$ | 2 | - | 2 | 61.29$_{\pm2.00}$ | 2 | - | 2 | 45.92$_{\pm2.00}$ | 2 | - |
| RM-SMA | 2 | 84.86$_{\pm1.50}$ | 2 | - | 2 | 80.26$_{\pm1.50}$ | 2 | - | 2 | 80.63$_{\pm2.00}$ | 2 | - | 2 | 61.45$_{\pm2.00}$ | 2 | - | 2 | 44.76$_{\pm2.00}$ | 2 | - |
| SMA | 1 | 97.08$_{\pm0.40}$ | 1 | 0 | 1 | 97.43$_{\pm0.40}$ | 1 | 0 | 1 | 93.26$_{\pm0.30}$ | 1 | 0 | 1 | 64.70$_{\pm0.40}$ | 1 | 0 | 1 | 50.71$_{\pm0.30}$ | 1 | 0 |
| **M-NCN** (Ours) | 50 | 95.40$_{\pm0.80}$ | 16 | 0.78 | 50 | 97.01$_{\pm0.50}$ | 23 | 0.80 | 50 | 90.42$_{\pm0.50}$ | 15 | 0.74 | 100 | 64.71$_{\pm0.70}$ | 33 | 0.81 | 100 | 61.12$_{\pm0.60}$ | 43 | 0.86 |
| DM-NCN | 2 | 85.52$_{\pm1.80}$ | 2 | - | 2 | 81.20$_{\pm1.20}$ | 2 | - | 2 | 81.34$_{\pm2.00}$ | 2 | - | 2 | 61.75$_{\pm2.00}$ | 2 | - | 2 | 57.46$_{\pm2.00}$ | 2 | - |
| KM-NCN | 2 | 83.86$_{\pm1.80}$ | 2 | - | 2 | 80.24$_{\pm1.20}$ | 2 | - | 2 | 81.15$_{\pm2.00}$ | 2 | - | 2 | 61.42$_{\pm2.00}$ | 2 | - | 2 | 57.81$_{\pm2.00}$ | 2 | - |
| RM-NCN | 2 | 89.73$_{\pm1.80}$ | 2 | - | 2 | 83.52$_{\pm1.20}$ | 2 | - | 2 | 83.10$_{\pm2.00}$ | 2 | - | 2 | 61.19$_{\pm2.00}$ | 2 | - | 2 | 57.48$_{\pm2.00}$ | 2 | - |
| NCN | 1 | 96.38$_{\pm0.40}$ | 1 | 0 | 1 | 97.20$_{\pm0.40}$ | 1 | 0 | 1 | 92.47$_{\pm0.50}$ | 1 | 0 | 1 | 64.75$_{\pm0.85}$ | 1 | 0 | 1 | 61.19$_{\pm0.87}$ | 1 | 0 |
| **M-LPFm** (Ours) | 50 | 95.43$_{\pm0.60}$ | 15 | 0.78 | 50 | 97.30$_{\pm0.60}$ | 23 | 0.80 | 50 | 92.80$_{\pm0.90}$ | 15 | 0.74 | 100 | 68.10$_{\pm0.70}$ | 33 | 0.81 | 100 | 63.39$_{\pm0.60}$ | 42 | 0.86 |
| DM-LPFm | 2 | 86.26$_{\pm1.50}$ | 2 | - | 2 | 81.18$_{\pm1.00}$ | 2 | - | 2 | 81.16$_{\pm2.00}$ | 2 | - | 2 | 62.00$_{\pm2.00}$ | 2 | - | 2 | 59.08$_{\pm2.00}$ | 2 | - |
| KM-LPFm | 2 | 85.58$_{\pm1.50}$ | 2 | - | 2 | 80.85$_{\pm1.80}$ | 2 | - | 2 | 80.45$_{\pm2.00}$ | 2 | - | 2 | 61.35$_{\pm2.00}$ | 2 | - | 2 | 59.10$_{\pm2.00}$ | 2 | - |
| RM-LPFm | 2 | 89.16$_{\pm1.60}$ | 2 | - | 2 | 83.00$_{\pm1.50}$ | 2 | - | 2 | 83.60$_{\pm2.00}$ | 2 | - | 2 | 62.14$_{\pm2.00}$ | 2 | - | 2 | 59.34$_{\pm2.00}$ | 2 | - |
| LPFm | 1 | 96.51$_{\pm0.40}$ | 1 | 0 | 1 | 97.39$_{\pm0.40}$ | 1 | 0 | 1 | 93.52$_{\pm0.30}$ | 1 | 0 | 1 | 68.14$_{\pm0.40}$ | 1 | 0 | 1 | 63.48$_{\pm0.30}$ | 1 | 0 |

the performance (AUC/HR@50/HR@100) and speedup ratio (SR) across different baselines.

In general, MIMO-LP achieves more than a **14–44**× acceleration in training and inference speed over standard one-to-one LP pipelines across different datasets and backbone models, without compromising accuracy. More importantly, due to its ability to multiplex the message-passing process inherent to many graph learning models, the speedup ratio (SR) of MIMO-LP consistently surpasses that of previous data multiplexing frameworks, including DM, KM, and RM. Table 1 further demonstrates that a higher ERR correlates with greater speedup across all five datasets for MIMO-LP, indicating that redundant message-passing among subgraphs is effectively eliminated during both training and inference. Notably, only MIMO-LP achieves a measurable ERR, since other baselines do not eliminate redundant message-passing operations. We further reports the breakdown of model training and inference time in Appendix E.5.

### 5.3. Impact of Multiplexing Count

To further evaluate the impact of the multiplexing count ($n$) on both acceleration and predictive accuracy, we vary its value within each batch. The results for acceleration are reported in Fig. 3, while predictive accuracy results are shown in Fig. 4. As shown in Fig. 3, increasing $n$ within a batch leads to higher acceleration, since more redundant message-passing operations are deduplicated within the union graph. In the meantime, Fig. 4 shows that MIMO-LP incurs degra-

dation of predictive accuracy when $n$ becomes excessive across all backbone models and datasets.

### 5.4. Ablation Studies

To evaluate the effectiveness of the proposed components in MIMO-LP, we design seven ablation variants based on M-SEAL: (1) a variant without the union graph (*w/o union graph*), which directly superposes the contextualized node feature vectors without considering topological information, (2) a variant without Gaussian noise matrix (*w/o $\phi$*), which directly superposes the subgraph feature vectors, (3) a variant without demultiplexing (*w/o de-MUX*), which performs prediction with the aggregated output representations, (4) a variant (*w/o clustering*) that constructs the union graph by randomly grouping subgraphs without the clustering method, (5) a variant (*w/o random mask*) that removes the random mask component, and (6) a variant (*w/o PReLU*) that remains the default activation function, (7) a variant (*w/o $\mathcal{L}_O$*) that removes the regularization term $\mathcal{L}_O$. The results are presented in Table 2. We find out that while all of the mechanisms contribute to the performance, union graph and clustering contributes the most to accuracy and SR, respectively. Appendix E.2 provides results on additional datasets and a detailed discussion.

## 6. Conclusion

In this paper, we propose MIMO-LP, a novel *Multi-Input Multi-Output* (MIMO) framework designed to accel-

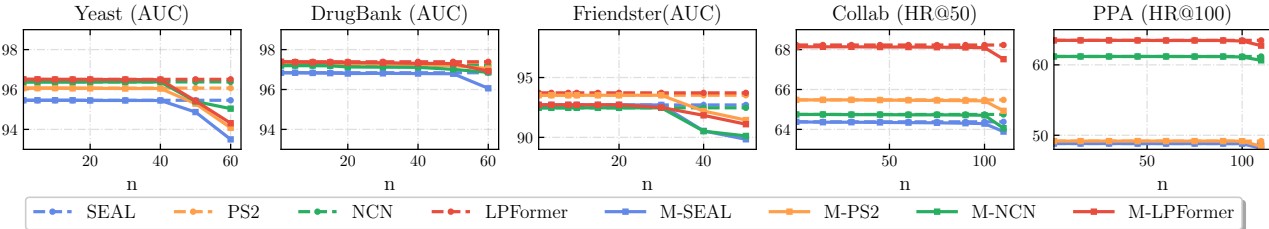

*Figure 3.* Speedup of our multiplexing models with varying multiplexing counts ($n$) on five datasets.

*Figure 4.* Performance of our multiplexing models with varying multiplexing counts ($n$) on five datasets.

*Table 2.* Ablation studies on MIMO-LP.

| Variants | Datasets | | | |
| | Yeast | | DrugBank | |
| | AUC | SR | AUC | SR |
|---|---|---|---|---|
| **M-SEAL** | **94.87**$_{\pm 0.50}$ | 16 | **96.79**$_{\pm 0.80}$ | 25 |
| w/o union graph | 69.80$_{\pm 1.00}$(**-25.07**) | 34 | 70.34$_{\pm 1.30}$(**-26.45**) | 42 |
| w/o $\phi$ | 91.26$_{\pm 0.50}$(**-3.61**) | 16 | 93.85$_{\pm 0.80}$(**-2.94**) | 25 |
| w/o de-MUX | 90.34$_{\pm 0.40}$(**-4.53**) | 16 | 93.07$_{\pm 0.50}$(**-3.72**) | 25 |
| w/o clustering | 95.26$_{\pm 0.50}$ | 12(**-4**) | 96.80$_{\pm 0.30}$ | 20(**-5**) |
| w/o random mask | 89.62$_{\pm 0.70}$(**-5.25**) | 16 | 94.26$_{\pm 0.70}$(**-2.53**) | 25 |
| w/o PReLU | 92.43$_{\pm 0.50}$(**-2.44**) | 16 | 94.97$_{\pm 0.70}$(**-1.82**) | 25 |
| w/o $\mathcal{L}_O$ | 93.15$_{\pm 0.50}$(**-1.72**) | 16 | 95.47$_{\pm 0.70}$(**-1.32**) | 25 |

erate subgraph-based link prediction (LP) methods through message-passing multiplexing. Extensive experiments demonstrate that MIMO-LP achieves up to **44× speedup** in both training and inference, while maintaining comparable predictive performance to state-of-the-art subgraph-based LP methods. In future work, we plan to extend MIMO-LP to dynamic and heterogeneous graph scenarios and explore its applicability to broader graph learning tasks to further improve scalability and generalization.

## Acknowledgement

We thank the anonymous reviewers for their constructive comments and suggestions. This work is supported in part by the National Natural Science Foundation of China (NSFC) under Grant Nos. 62232004, 92467205, 62502086, and U25A6024, the Natural Science Foundation of Jiangsu Province under Grant No. BK20251295, the Start-up Research Fund of Southeast University under Grant No. RF1028624178, the Jiangsu Provincial Key Laboratory of Network and Information Security under Grant No. BM2003201, the Key Laboratory of Computer Network and Information Integration of Ministry of Education of China under Grant No. 93K-9, and the Collaborative Innovation Center of Novel Software Technology and Industrialization. We also acknowledge the support of the Big Data Computing Center of Southeast University.

## Impact Statement

This paper presents work whose goal is to advance the field of machine learning. There are many potential societal consequences of our work, none of which we feel must be specifically highlighted here.

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

# A. Subgraph Batching

## A.1. Group with Clustering

Intuitively, a higher $\mathrm{ERR}$ indicates that more redundant computations can be eliminated. The maximum $\mathrm{ERR}$ would occur in the ideal case where all subgraphs are merged into a single group, i.e., $G_{\mathcal{B}_k} = G$. However, due to the potential interference among multiplexed signals (see Section 5.3), the maximum number of multiplexed queries per batch is limited. Therefore, the subgraphs corresponding to different query pairs must be divided into $K$ disjoint groups, where $K$ is determined by the optimal multiplexing number $n$ (see detailed discussion in Section A.2) allowed per batch. This setting makes the subgraph batching objective analogous to a variant of the bin packing problem.

Since this optimization problem is NP-complete, we approach it using a clustering-based approximation. We first construct a metric function whose clustering outcome aligns with the optimization objective in Eq. 5. Because clustering similarity is computed pairwise, this metric measures the pairwise edge reduction ratio (ERR) between two subgraphs. Specifically, for any two subgraph edge sets $\mathcal{E}_{G_m}$ and $\mathcal{E}_{G_n}$, we define:

$$\mathrm{ERR}(\mathcal{E}_{G_m}, \mathcal{E}_{G_n}) = \frac{|\mathcal{E}_{G_m}| + |\mathcal{E}_{G_n}| - |\mathcal{E}_{G_m} \cup \mathcal{E}_{G_n}|}{|\mathcal{E}_{G_m}| + |\mathcal{E}_{G_n}|}. \tag{16}$$

However, direct computation of Eq. 16 is computationally expensive due to repeated cardinality and intersection operations. Alternatively, we show that maximizing Eq. 16 is equivalent to maximizing the Jaccard Index between $\mathcal{E}_{G_m}$ and $\mathcal{E}_{G_n}$, owing to their monotonic increasing relationship, as formalized by Theorem A.1:

**Theorem A.1.** *Given the edge sets $\mathcal{E}_{G_m}$ and $\mathcal{E}_{G_n}$ of two subgraphs, the Jaccard Index $J(\mathcal{E}_{G_m}, \mathcal{E}_{G_n})$ increases monotonically with the pair-wise Redundant Reduction $\mathrm{ERR}(\mathcal{E}_{G_m}, \mathcal{E}_{G_n})$, where*

$$\mathcal{J}(\mathcal{E}_{G_m}, \mathcal{E}_{G_n}) = \frac{|\mathcal{E}_{G_m} \cap \mathcal{E}_{G_n}|}{|\mathcal{E}_{G_m} \cup \mathcal{E}_{G_n}|} \tag{17}$$

*Proof.* According to the operations between sets, Eq. 16 can be rewritten as

$$\begin{aligned}
\frac{|\mathcal{E}_{G_m}| + |\mathcal{E}_{G_n}| - |\mathcal{E}_{G_m} \cup \mathcal{E}_{G_n}|}{|\mathcal{E}_{G_m}| + |\mathcal{E}_{G_n}|} &= \frac{|\mathcal{E}_{G_m} \cap \mathcal{E}_{G_n}|}{|\mathcal{E}_{G_m} \cap \mathcal{E}_{G_n}| + |\mathcal{E}_{G_m} \cup \mathcal{E}_{G_n}|} \\
&= 1 - \frac{1}{1 + \frac{|\mathcal{E}_{G_m} \cap \mathcal{E}_{G_n}|}{|\mathcal{E}_{G_m} \cup \mathcal{E}_{G_n}|}} \\
&= 1 - \frac{1}{1 + \mathcal{J}(\mathcal{E}_{G_m}, \mathcal{E}_{G_n})}.
\end{aligned}$$

$\square$

As a result, by applying a clustering algorithm (e.g., $k$-means) using the Jaccard Index as the similarity metric over the edge sets of subgraphs, we can efficiently derive a batching assignment that achieves near-optimal $\mathrm{ERR}$.

## A.2. Selection of Multiplexing Query Count $n$

We define the optimal multiplexing count $n^\star$ as the maximum $n$ that does not cause a reduction in predictive accuracy. In this section, we present a data-driven approach to determine $n^\star$.

The proposed method is motivated by the observation that interference arises when messages from different subgraphs propagate through shared edges. Hence, the larger the number of shared edges, the more severe the interference. Previous studies (Di Giovanni et al., 2023; Topping et al., 2022) show that the degree of edge sharing among subgraphs in a given graph can be quantified by the *Cheeger constant $h(G)$*:

$$h(G) = \min_{G_i \in \mathcal{G}} \frac{|\{(u, v) \in \mathcal{E} \mid u \in \mathcal{V}_{G_i}, v \in \mathcal{V} \setminus \mathcal{V}_{G_i}\}|}{\min\{|\mathcal{V}_{G_i}|, |\mathcal{V} \setminus \mathcal{V}_{G_i}|\}}, \tag{18}$$

where $G = (\mathcal{V}, \mathcal{E})$ denotes the given graph, and $\{G_i \mid G_i \in \mathcal{G}\}$ represents the set of extracted subgraphs. The numerator in Eq. 18 measures the number of edges shared between a subgraph $G_i$ and the remaining graph $\mathcal{G} \setminus G_i$, while the denominator

normalizes this by the smaller of the two vertex set sizes. Intuitively, $h(G)$ measures how many edges must be removed to partition $G$ into two large components, where a larger $h(G)$ implies that the graph is more densely connected. Therefore, $h(G)$ effectively quantifies the extent of edge sharing between different vertex sets (i.e., subgraphs).

According to the Cheeger inequality (Chung, 1996), $h(G)$ is bounded by spectral properties of the graph:

$$\frac{\lambda_2}{2} \le h(G) \le \sqrt{2\lambda_2}, \tag{19}$$

where $\lambda_2 \in (0, 2)$ denotes the second-smallest eigenvalue of the Laplacian matrix of $G$. This eigenvalue can be computed efficiently using eigenvalue decomposition techniques such as the Lanczos algorithm or LOBPCG method (Knyazev, 2001). A larger $\lambda_2$ indicates stronger graph connectivity and thus more shared edges, implying a smaller desired multiplexing count $n$ to limit signal interference during propagation, and vice versa. Based on this insight, we establish a linear relation between the spectral measure and the optimal multiplexing count $n^\star$:

$$n = \lceil \alpha\lambda_2 + b \rceil, \tag{20}$$

where $\lceil \cdot \rceil$ denotes the rounding operation to the nearest integer, $\alpha$ is a scaling factor, and $b$ is a bias term. We empirically obtain pairs of $(h(G), n^\star)$ by executing MIMO-LP across various benchmark datasets, and estimate $\alpha = 43$ and $b = 0$ through linear regression. Finally, Algorithm 1 summarizes the above procedure for determining the optimal $n^\star$ given $G$.

---

**Algorithm 1** Selection of Multiplexing Query Count $n$

---

1: **Input:** Graph $G$; scaling factor $\alpha$; bias term $b$ estimated from Eq. 20.
2: Extract the adjacency matrix $\mathbf{A}$ and degree matrix $\mathbf{D}$ from $G$.
3: Compute the unnormalized Laplacian: $\mathbf{L} = \mathbf{D} - \mathbf{A}$.
4: Estimate the eigenvalue spectrum $\boldsymbol{\lambda}$ using the Lanczos or LOBPCG method: $\boldsymbol{\lambda} = \texttt{Lanczos}(\mathbf{L})$ **or** $\texttt{LOBPCG}(\mathbf{L})$.
5: Obtain the second smallest eigenvalue (Fiedler value): $\lambda_2 = \texttt{unique\_sort}(\boldsymbol{\lambda})[1]$.
6: Compute the multiplexing query count: $n = \alpha\lambda_2 + b$.
7: **Output:** Multiplexing query count $n$.

---

# B. Interference Mitigation

## B.1. Superposition Theory

In this section, we theoretically demonstrate why the message-passing processes from multiple subgraphs be multiplexed by MIMO-LP in a single forward pass. We first introduce Assumptions B.1 and B.2, upon which our theoretical demonstration is built.

**Assumption B.1** (Quasi-orthogonality). Let $\{\boldsymbol{\Phi}_{G_i}\}_{G_i \in G_{\mathcal{B}_k}}$ be $n$ Gaussian noise vectors and let $\odot$ denote the Hadamard product. Given a set of subgraph representations $\{\widetilde{\mathbf{X}}_{G_i}\}_{G_i \in G_{\mathcal{B}_k}}$, the operation $\boldsymbol{\Phi}_{G_i} \odot \widetilde{\mathbf{X}}_{G_i}$ maps each representation $\widetilde{\mathbf{X}}_{G_i}$ into a distinct orthogonal subspace. Therefore, the resulting terms for distinct indices, such as $\boldsymbol{\Phi}_{G_i} \odot \widetilde{\mathbf{X}}_{G_i}$ and $\boldsymbol{\Phi}_{G_j} \odot \widetilde{\mathbf{X}}_{G_j}$ for $G_i \ne G_j$ exhibit quasi-orthogonality.

Assumption B.1 implies that the transformations $\{\boldsymbol{\Phi}_{G_i}\}_{G_i \in G_{\mathcal{B}_k}}$ map instances $\{\widetilde{\mathbf{X}}_{G_i}\}_{G_i \in G_{\mathcal{B}_k}}$ at distinct indices to distinguishable regions. This assumption is supported by studies (Menet et al., 2024; Murahari et al., 2022).

**Assumption B.2** (Separability). Consider a superposition of $n$ mutually orthogonal terms $\{\boldsymbol{\Phi}_{G_i} \odot \widetilde{\mathbf{X}}_{G_i}\}_{G_i \in G_{\mathcal{B}_k}}$:

$$\mathbf{S} = \sum_{G_i \in G_{\mathcal{B}_k}} \left( \boldsymbol{\Phi}_{G_i} \odot \widetilde{\mathbf{X}}_{G_i} \right). \tag{21}$$

If the superposed signal is processed by a neural network $\mathcal{F}()$ with the aim of retrieving independent outputs via an optimized MLP transformation:

$$\mathcal{F}(\widetilde{\mathbf{X}}_{G_i}) = \text{MLP}(\mathbf{P}_{G_i} \oplus \mathcal{F}(\mathbf{S})) \tag{22}$$

where $\mathbf{P}_{G_i}$ is the corresponding retrieving prefix and $\oplus$ denotes concatenation, then $\mathcal{F}()$ must satisfy **functional additivity**. That is,

$$\mathcal{F}(\mathbf{S}) = \sum_{i=1}^{n} \mathcal{F}(\boldsymbol{\Phi}_{G_i} \odot \widetilde{\mathbf{X}}_{G_i}) \tag{23}$$

As evidenced by numerous studies (Menet et al., 2024; Murahari et al., 2022; Su et al., 2023), Assumption B.2 is a prevalent concept in the study of data multiplex. Assumption B.2 states that if the outputs of $\mathcal{F}(\cdot)$ remain in a superposed state, i.e., they occupy distinguishable regions within the same representation space, then they can be separated without loss.

Based on Assumptions B.1 and B.2, as long as MIMO-LP preserves the superposition of multiple inputs within $\mathcal{F}_{\text{l-enc}}$, the message-passing processes of multiple subgraphs can be multiplexed. This property can be formalized as the **functional additivity** of $\mathcal{F}_{\text{l-enc}}$, as stated in Proposition B.3.

**Proposition B.3** (Functional Additivity). *Given a graph encoder $\mathcal{F}_{\text{l-enc}}$ and a union graph $G_{\mathcal{B}_k}$, applying $\mathcal{F}_{\text{l-enc}}$ to superimposed inputs on $G_{\mathcal{B}_k}$ is approximately equivalent to processing these inputs independently on each subgraph, i.e.,*

$$
\mathbf{H}_{G_{\mathcal{B}_k}} \triangleq \mathcal{F}_{\text{l-enc}}\Big(\mathcal{F}_{\text{M}}\Big(\widetilde{\mathbf{X}}_{G_1}, \ldots, \widetilde{\mathbf{X}}_{G_n}\Big), G_{\mathcal{B}_k}\Big)
$$

$$
\approx \mathbf{D}^{-1} \sum_{G_i \in \mathcal{B}_k} \mathcal{F}_{\text{l-enc}}\Big(\boldsymbol{\Phi}_{G_i} \odot \widetilde{\mathbf{X}}_{G_i}, G_{\mathcal{B}_k}\Big) \tag{24}
$$

$$
\approx \mathbf{D}^{-1} \sum_{G_i \in \mathcal{B}_k} \mathcal{F}_{\text{l-enc}}\Big(\boldsymbol{\Phi}_{G_i} \odot \widetilde{\mathbf{X}}_{G_i}, G_i\Big). \tag{25}
$$

To prove Proposition B.3, we establish Eqs. 24 and 25 via Theorems B.4 and B.5, respectively.

**Theorem B.4** (Semi-linearity of $\mathcal{F}_{\text{l-enc}}$). *Given a nonlinear encoder $\mathcal{F}_{\text{l-enc}}$ parameterized by $\mathbf{W}$ with a semi-linear Parametric Rectified Linear Unit (PReLU (He et al., 2015)) activation controlled by parameter $b$, there exist parameters $\mathbf{W}, b$ such that $\mathcal{F}_{\text{l-enc}}$ satisfies additivity for any input $\mathbf{X}_{G_{\mathcal{B}_k}} = \widetilde{\mathbf{X}}_{G_i} + \widetilde{\mathbf{X}}_{G_j}$ on the union graph $G_{\mathcal{B}_k}$:*

$$
\mathcal{F}_{\text{l-enc}}(\mathbf{X}_{G_{\mathcal{B}_k}}, G_{\mathcal{B}_k}) \approx \mathcal{F}_{\text{l-enc}}(\widetilde{\mathbf{X}}_{G_i}, G_{\mathcal{B}_k}) + \mathcal{F}_{\text{l-enc}}(\widetilde{\mathbf{X}}_{G_j}, G_{\mathcal{B}_k}). \tag{26}
$$

*Proof.* The encoder $\mathcal{F}_{\text{l-enc}}$ can be represented in the form of a Message Passing Neural Network (MPNN), consisting of two steps. The first step is message passing:

$$
\mathbf{M}^{(l)} = \mathbf{A} \cdot \mathbf{H}^{(l-1)} \cdot \mathbf{W}^{(l)}, \tag{27}
$$

where $\mathbf{A}$ is the adjacency matrix of the union graph, $\mathbf{H}^{(l-1)}$ is the representation from the previous layer, and $\mathbf{W}^{(l)}$ is a learnable weight matrix. Since this step involves inner products, it preserves linearity according to inner-product preservation property (Menet et al., 2024). The second step is feature update:

$$
\mathbf{H}^{(l)} = \sigma(\mathbf{H}^{(l-1)} + \mathbf{M}^{(l)}), \tag{28}
$$

where $\sigma$ denotes the activation function (typically ReLU). As shown in (Murahari et al., 2022), the attention mechanism can be approximated as linear. Thus, the main source of non-linearity in an MPNN arises from $\sigma(\cdot)$. To preserve the linear feature composition essential for multiplexing, we replace ReLU with the semi-linear PReLU function:

$$
\text{PReLU}_b(x) = \max(x, 0) + b \cdot \min(x, 0), \tag{29}
$$

where the learnable parameter $b \in [-1, 1]$ controls the degree of linearity. When $b = 1$, the mapping is fully linear. Therefore, for sufficiently linear behavior, $\mathcal{F}_{\text{l-enc}}$ satisfies Eq. 26. $\square$

Theorem B.4 states that on a union graph, applying $\mathcal{F}_{\text{l-enc}}$ to the superposition of features from two subgraphs is equivalent to applying $\mathcal{F}_{\text{l-enc}}$ to each feature set separately.

Now, we demonstrate that applying $\mathcal{F}_{\text{l-enc}}$ to the union graph is equivalent to applying it to the subgraphs in Theorem B.5.

**Theorem B.5.** *Given an $h$-hop enclosing subgraph $G_i$ for a query and its corresponding union graph $G_{\mathcal{B}_k}$ (constructed by merging overlapping subgraphs), the heuristics computed on $G_i$ and $G_{\mathcal{B}_k}$ tend to converge as $h$ increases, i.e.,*

$$
\mathcal{F}_{\text{l-enc}}(\widetilde{\mathbf{X}}_{G_i}, G_{\mathcal{B}_k}) \approx \mathcal{F}_{\text{l-enc}}(\widetilde{\mathbf{X}}_{G_i}, G_i). \tag{30}
$$

Before proving Theorem B.5, we begin by recalling the Definition B.6 and Lemmas B.7 and B.8.

**Definition B.6** (Enclosing Subgraph (Zhang & Chen, 2018))**.** For a graph $G = (\mathcal{V}, \mathcal{E})$ and nodes $u, v \in \mathcal{V}$, the $h$-hop enclosing subgraph for $(u, v)$ is defined as the subgraph $G_{u,v}^h$ induced by the nodes $\{i \mid d(i, u) \leq h \text{ or } d(i, v) \leq h\}$, where $d(\cdot, \cdot)$ denotes hop distance.

**Lemma B.7** (Heuristic Decay Theorem (Zhang & Chen, 2018))**.** *Any $h$-order heuristic for $(u, v)$ can be accurately computed from $G_{u,v}^h$ in a $\gamma$-decaying form:*

$$\mathbf{H}_{G_{u,v}^h} = \eta \sum_{l=1}^{h} \gamma^l \mathcal{F}_{\text{l-enc}}^{(l)}(\mathbf{X}_{G_{u,v}^h}, G_{u,v}^h), \tag{31}$$

*where $\gamma \in (0, 1)$ is the decay factor, $\eta$ is a positive constant, and $\mathcal{F}_{\text{l-enc}}^{(l)}$ is a nonnegative function.*

*Proof.* See the proof of Theorem 2 in (Zhang & Chen, 2018).

**Lemma B.8.** *Let $G_i$ be the $h$-hop enclosing subgraph of $(u, v)$. The union graph $G_{\mathcal{B}_k}$ is obtained by merging $G_i$ with another overlapping enclosing subgraph $G_j$. Then any node $x$ in $G_{\mathcal{B}_k}$ satisfies:*

$$\{x \mid d(x, u) \leq (h + \Delta_u) \text{ or } d(x, v) \leq (h + \Delta_v)\},$$

*where $\Delta_u, \Delta_v \geq 0$ and $\Delta_u, \Delta_v \in \mathbb{Z}_+$.*

*Proof.* Let $\mathcal{V}_{G_i}$ and $\mathcal{V}_{G_j}$ denote the node sets of subgraphs $G_i$ and $G_j$, respectively. We discuss it in three cases:

1. When any node $x \in (\mathcal{V}_{G_j} - \mathcal{V}_{G_i})$, the hop $d(x, u) > h$ or $d(x, v) > h$. Because any node $x \in \mathcal{V}_{G_i}$ satisfies the hop $\{x \mid d(x, u) \leq h \text{ or } d(x, v) \leq h\}$ in the definition of enclosed subgraphs B.6. When node $x \notin \mathcal{V}_{G_i}$, the hop $\{x \mid d(x, u) > h \text{ or } d(x, v) > h\}$. In this case, the hop increment $\Delta_u, \Delta_v > 0$.

2. When any node $x \in (\mathcal{V}_{G_i} - \mathcal{V}_{G_j})$, the hop $d(x, u) \leq (h + \Delta_u)$ or $d(x, v) \leq (h + \Delta_v)$. Let the node set $\mathcal{P}$ denotes all the nodes in the path from $x$ to $u$ (or $v$). If $\mathcal{P} \subset \mathcal{V}_{G_i}$, the hop is not changed, i.e., $\Delta_u, \Delta_v = 0$, else the hop increases, i.e., $\Delta_u, \Delta_v > 0$.

3. When any node $x \in (\mathcal{V}_{G_i} \cap \mathcal{V}_{G_j})$, the hop $d(x, u) \leq (h + \Delta_u)$ or $d(x, v) \leq (h + \Delta_v)$. Case 3 is analogous to Case 2. We can discuss whether the increment $\Delta_u, \Delta_v = 0$ through the node set $\mathcal{P}$.

According to the analysis of the three cases above, any node $x$ on the union graph $G_{\mathcal{B}_k}$ satisfies: $\{x \mid d(x, u) \leq (h + \Delta_u) \text{ or } d(x, v) \leq (h + \Delta_v)\}$, where $\Delta_u, \Delta_v \geq 0$ and $\Delta_u, \Delta_v \in \mathbb{Z}_+$.

We now proceed to prove Theorem B.5.

*Proof.* For the query node pair $(u, v)$, we re-express $G_i$ and $G_{\mathcal{B}_k}$ using the definition of enclosing subgraphs in Definition B.6. Specifically, the subgraph $G_i$ can be represented as $G_{u,v}^h$, which is the subgraph induced from $G$ by the set of nodes $\{x \mid d(x, u) \leq h \text{ or } d(x, v) \leq h\}$. According to Lemma B.8, the corresponding union graph $G_{\mathcal{B}_k}$ can be expressed as $G_{u,v}^{h+\Delta}$, induced from $G$ by the set of nodes $\{x \mid d(x, u) \leq (h + \Delta_u) \text{ or } d(x, v) \leq (h + \Delta_v)\}$. Consequently, the approximation error $\epsilon$ between $\mathbf{H}_{G_{u,v}^h}$ and $\mathbf{H}_{G_{u,v}^{h+\Delta}}$ can be derived based on Lemma B.7 as follows:

$$\begin{aligned} \epsilon &= \left| \mathcal{F}_{\text{l-enc}}(\widetilde{\mathbf{X}}_{G_i}, G_{\mathcal{B}_k}) - \mathcal{F}_{\text{l-enc}}(\widetilde{\mathbf{X}}_{G_i}, G_i) \right| \\ &= \left| \mathbf{H}_{G_{u,v}^{h+\Delta}} - \mathbf{H}_{G_{u,v}^h} \right| \\ &= \eta \sum_{l=h}^{h+\Delta} \gamma^l \mathcal{F}_{\text{l-enc}}^{(l)}(\widetilde{\mathbf{X}}_{G_{u,v}^{h+\Delta}}, G_{u,v}^{h+\Delta}). \end{aligned} \tag{32}$$

In Eq. 32, the approximation error $\epsilon$ decays at least exponentially with respect to $h$, since $\gamma \in (0, 1)$. Therefore, when $h$ is sufficiently large, this approximation error becomes negligible, establishing the asymptotic equivalence between $\mathcal{F}_{\text{l-enc}}(\widetilde{\mathbf{X}}_{G_i}, G_{\mathcal{B}_k})$ and $\mathcal{F}_{\text{l-enc}}(\widetilde{\mathbf{X}}_{G_i}, G_i)$. $\qquad \square$

Based on Theorems B.4 and B.5, we proceed to the proof of Proposition B.3.

*Proof.* In Eq. 24, the superposition of $n$ subgraph features, i.e.,

$$\mathcal{F}_{\mathrm{M}}\left(\widetilde{\mathbf{X}}_{G_1}, \ldots, \widetilde{\mathbf{X}}_{G_n}\right) = \mathbf{D}^{-1} \sum_{G_i \in \mathcal{B}_k} \left(\boldsymbol{\Phi}_{G_i} \odot \widetilde{\mathbf{X}}_{G_i}\right),$$

is processed by the graph encoder $\mathcal{F}_{\text{l-enc}}()$. Since $\mathcal{F}_{\text{l-enc}}()$ is approximately linear, as demonstrated in Theorem B.4, these superposed subgraphs are transformed by a single application of encoder, enabling all subgraphs to be processed jointly in a superposition. Therefore, Eq. 24 holds. Based on Theorem B.5, the encoding operation on the union graph is equivalent to applying the graph encoder to its constituent subgraphs individually. Consequently, Eq. 25 holds. $\square$

### B.2. A theoretical bound to quantify performance degradation from multiplexing

We briefly derive the approximation error bound of the multiplexing based on the Superposition Theory in Appendix B.1.

To recap, MIMO-LP constructs a union graph by merging subgraphs and multiplexes feature information over overlapping regions. The approximation error for each individual subgraph representation consists of two components: the linearity error $\epsilon_{\text{lin}}$ and the topology error $\epsilon_{\text{top}}$.

**Linearity error $\epsilon_{\text{lin}}$.** As established in **Thm.B.4**, signals propagated through the semi-linear encoder $\mathcal{F}_{\text{l-enc}}$ can be approximately superimposed due to its near-linearity. The approximation error arises from replacing ReLU with PReLU:

$$\text{PReLU}_b(x) = \max(x, 0) + b \cdot \min(x, 0),$$

where $b \in [-1, 1]$ controls the degree of linearity. When $b = 1$, the mapping is fully linear, yielding $\epsilon_{\text{lin}} = \mathcal{O}(|1 - b|)$.

**Topology error $\epsilon_{\text{top}}$.** As established in Thm. B.5, the effect of the hop increment $\Delta$ induced by the union graph on the representation of the original subgraph can be quantified by $\epsilon_{\text{top}}$. Specifically, following the decay theory in Thm. B.7, the representation of an $h$-hop subgraph obtained by the semi-linear encoder is given by $\sum_{l=1}^{h} \gamma^l \mathcal{F}_{\text{l-enc}}^{(l)}$, where $h$ denotes the hop radius from the query node pair. Accordingly, the topology error induced by using the union graph instead of the original subgraph is

$$\epsilon_{\text{top}} = \left| \sum_{l=1}^{h+\Delta} \gamma^l \mathcal{F}_{\text{l-enc}}^{(l)} - \sum_{l=1}^{h} \gamma^l \mathcal{F}_{\text{l-enc}}^{(l)} \right| = \sum_{l=h}^{h+\Delta} \gamma^l \mathcal{F}_{\text{l-enc}}^{(l)},$$

where $\gamma \in (0, 1)$ is the decay factor and $l$ denotes the hop distance. Notably, $\epsilon_{\text{top}}$ decays exponentially as $h$ increases.

**Total error bound.** For multiplexing over subgraphs, the total error is bounded as: $\epsilon_{\text{total}} = n\epsilon_{\text{subgraph}}$, where $\epsilon_{\text{subgraph}} = \epsilon_{\text{lin}} + \epsilon_{\text{top}}$.

### B.3. Noise from the Exponentially Expanding Neighborhood

**Assumption B.9** (Over-squashing). Let $\mathbf{x}_i$ denote the initial feature of node $i$. The graph encoder $\mathcal{F}_{\text{l-enc}}$ employs a message aggregation mechanism on each node. Specifically, after a single aggregation step, node $i$ acquires a representation of its first-order neighborhood, denoted as

$$\mathbf{h}_i^{(1)} = aggregator^{(1)}(\{\mathbf{x}_j\}|j \in \mathcal{N}(i)) \tag{33}$$

where $\mathcal{N}(i)$ represents the set of immediate neighbors of node $i$. By performing this aggregation recursively for $l$ iterations, node $i$ integrates information from all neighbors within its $l$-hop range. While expanding the receptive field enriches the semantic information within $\mathbf{h}_i^{(l)}$, the fixed dimensionality of the embedding vectors leads to the over-squashing of this neighborhood information.

The assumption of over-squashing is a prevalent concept in the domain of graph learning, as evidenced by many studies (Topping et al., 2022; Di Giovanni et al., 2023). In our framework, the union graph expands the neighborhood scope of each node, thereby potentially introducing the risk of information over-squashing. To alleviate information over-squashing, we propose a random masking mechanism to prune partial node information.

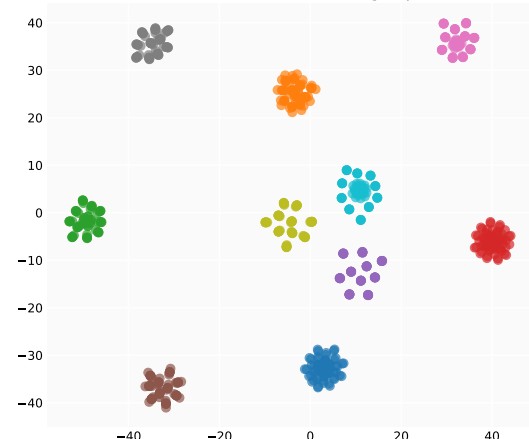

*Figure 5.* The visualization of demultiplexing representation.

## C. The Visualization of Demultiplexing Representation

To empirically validate Assumptions B.1 and B.2, we conduct a experiment to analyze whether our demultiplexing function $\mathcal{F}_{\text{DM}}$ can effectively dissociate multiple outputs from a single multiplexed representation. In our experimental setup, we randomly select 10 subgraphs for data multiplexing-based inference in M-SEAL on the Yeast dataset. After processing with the demultiplexing function, we used t-SNE to visualize the demultiplexed representations. In Fig. 5, 10 different colors represent 10 subgraphs respectively. We observe that nodes belonging to the same subgraph are clustered closely together, and this result indicates that the demultiplexing function can distinguish nodes from different subgraphs.

## D. Complexity Analysis

### D.1. Time Complexity Analysis

In this section, we analyze the computational overhead of MIMO-LP in comparison to the standard one-to-one subgraph-based LP pipeline.

Let $n$ denote the total number of subgraphs and $m$ denote the average number of edges per subgraph. Each subgraph is embedded into a $d$-dimensional representation. Recall that standard subgraph-based LP pipeline has to process $n$ subgraphs one by one, where the message-passing operations involved in each subgraph is proportional to $m$. Therefore, the time complexity of the standard one-to-one subgraph-based LP pipeline is $\mathcal{O}(mn)$.

For MIMO-LP, the primary computational overhead arises from three components: (1) subgraph batching, (2) union-graph construction (3) multiplexing, and (4) demultiplexing. During subgraph batching, MIMO-LP first employs $c$ MinHash functions to estimate pairwise similarity among subgraphs, incurring a computational cost of $\mathcal{O}(cmn)$. Subsequently, $t$ iterations of the $k$-means algorithm are performed to generate clusters, resulting in a complexity of $\mathcal{O}(tdn)$. Consequently, the total overhead for subgraph batching is $\mathcal{O}(tdn + cmn)$, which is of the same order as the baseline one-to-one subgraph-based LP methods and is negligible in practice (see Appendix E.5). Union graph construction yields a complexity of $\mathcal{O}(mn)$. During the multiplexing stage, MIMO-LP aggregates features from $n$ subgraphs along shared edges, incurring a computational cost of $\mathcal{O}((1 - \text{ERR})mn)$, where ERR denotes the edge reduction rate. Orthogonality regularization and semi-linear activation (PReLU) are applied within the GNN backbone and incur $\mathcal{O}(n)$. In the demultiplexing stage, MIMO-LP employs a $d_{\text{demux}}$-dimensional vector to recover individualized outputs from the aggregated $d$-dimensional hidden representation using a two-layer MLP. The computational complexity of this process is $\mathcal{O}((d_{\text{demux}} + d)d_{\text{hd}}n + d_{\text{hd}}d_{\text{class}}n)$, where $d_{\text{hd}}$ denotes the hidden dimension of the MLP and $d_{\text{class}}$ denotes the number of output classes. Overall, the total computational complexity of MIMO-LP is given by

$$\mathcal{O}\big(tdn + cmn + mn + (1 - \text{ERR})mn + n + (d_{\text{demux}} + d)d_{\text{hd}}n + d_{\text{hd}}d_{\text{class}}n\big). \tag{34}$$

Since $c$, $t$, $d$, $d_{\text{hd}}$, $d_{\text{demux}}$, and $d_{\text{class}}$ are constants and negligible compared to $m$ and $n$, Equation (34) indicates that the overall complexity of MIMO-LP remains $\mathcal{O}(mn)$, matching that of the baseline one-to-one subgraph-based LP methods.

Although MIMO-LP introduces a small additional overhead, it eliminates redundant computations on overlapping edges, effectively reducing the dominant cost to $\mathcal{O}((1 - \mathrm{ERR})mn)$. In the worst-case scenario where subgraphs are completely disjoint ($\mathrm{ERR} = 0$), the computational complexity of MIMO-LP degenerates to $\mathcal{O}(mn)$. However, across real-world datasets (including the datasets evaluated in this paper), ERR typically exceeds $70\%$, resulting in substantial computational savings.

### D.2. Space Complexity Analysis

Following the previous notation, we analyze the space complexity of MIMO-LP relative to the standard one-to-one subgraph-based LP pipeline.

Recall that the standard subgraph-based LP pipeline stores $n$ subgraphs independently. Since the number of nodes per subgraph reaches $m + 1$ (chain topology) in the worst case, the storage overhead scales with $m$. Consequently, the total storage overhead is $\mathcal{O}(mnd)$, where $d$ denotes the node feature dimension.

For MIMO-LP, the primary storage overhead arises from three components: (1) input subgraph, (2) demultiplexing index and (3) two-layer MLP network. Unlike the standard link prediction pipeline that processes each subgraph independently, MIMO-LP handles multiple subgraph samples via a union graph approach, incurring a storage overhead of $\mathcal{O}((1 - \mathrm{ERR})mnd)$. MIMO-LP assigns a $d_{\mathrm{demux}}$-dimensional demultiplexing prefix vector to each subgraph representation, incurring an overhead of $nd_{\mathrm{demux}}$. Additionally, MIMO-LP employs a two-layer MLP network to transform each independent subgraph representation, incurring an overhead of $(d_{\mathrm{demux}} + d)d_{\mathrm{hd}} + d_{\mathrm{hd}}d_{\mathrm{class}}$. Overall, the total storage complexity of MIMO-LP is given by

$$\mathcal{O}\big((1 - \mathrm{ERR})mnd + nd_{\mathrm{demux}} + (d_{\mathrm{demux}} + d)d_{\mathrm{hd}} + d_{\mathrm{hd}}d_{\mathrm{class}}\big). \tag{35}$$

Since $d$, $d_{\mathrm{hd}}$, $d_{\mathrm{demux}}$, and $d_{\mathrm{class}}$ are constants and negligible compared to $m$ and $n$, Equation (35) indicates that the overall space complexity of MIMO-LP remains $\mathcal{O}(mnd)$, matching that of the baseline one-to-one subgraph-based LP methods.

Although MIMO-LP introduces a small additional overhead, it eliminates redundant storage overhead on overlapping nodes, effectively reducing the dominant cost to $\mathcal{O}((1 - \mathrm{ERR})mnd)$. Notably, in real-world datasets (including the datasets evaluated in this paper), ERR typically exceeds $70\%$, resulting in significant reductions in storage overhead.

## E. Experiment

### E.1. Dataset Description

**Yeast** (Von Mering et al., 2002) and **PPA** (Hu et al., 2020) are two proteins-protein interaction networks, where each node represents the protein and each edge represents the interactions. **DrugBank** (Newman, 2006) is a drug-drug interaction network, where each node represents drug and each edge represents interaction types. Notably, DrugBank is a multi-classes datasets (i.e., 86 interaction types). **Friendster** (Yang & Leskovec, 2012) is a social network originating from the Friendster social platform. In this graph, nodes represent individual users, while edges denote the interactions between them. **Collab** (Hu et al., 2020) are two networks of collaboration between researchers, where each node represents the author and each edge represents the cooperative relationship between two authors. The statistics of the datasets are summarized in TABLE 3.

*Table 3.* Statistics of the 5 datasets evaluated in this section.

| Statistics | Yeast | DrugBank | Friendster | Collab | PPA |
|---|---|---|---|---|---|
| # of Vertices | 2,375 | 12,015 | 65,608,366 | 235,868 | 576,289 |
| # of Edges | 11,693 | 1,895,445 | 1,806,067,135 | 1,285,465 | 30,326,273 |

### E.2. Ablation Studies

To evaluate the effectiveness of the proposed components in MIMO-LP, we design five ablation variants based on M-SEAL: (1) a variant without the union graph (*w/o union graph*), which directly superposes the contextualized node feature vectors without considering topological information, (2) a variant without Gaussian noise matrix (*w/o φ*), which directly superposes the subgraph feature vectors, (3) a variant without demultiplexing (*w/o de-MUX*), which performs prediction with the

*Table 4.* Ablation studies on MIMO-LP.

| Variants | Datasets | | | | | | | | | |
| | Yeast | | DrugBank | | Friendster | | Collab | | PPA | |
| | AUC | SR | AUC | SR | AUC | SR | HR@50 | SR | HR@100 | SR |
|---|---|---|---|---|---|---|---|---|---|---|
| **M-SEAL** | **94.87**$_{\pm 0.50}$ | 16 | **96.79**$_{\pm 0.80}$ | 25 | **90.38**$_{\pm 0.50}$ | 15 | **64.29**$_{\pm 0.60}$ | 34 | **48.80**$_{\pm 0.50}$ | 44 |
| w/o union graph | 69.80$_{\pm 1.00}$(**-25.07**) | 34 | 70.34$_{\pm 1.30}$(**-26.45**) | 42 | 62.59$_{\pm 1.00}$(**-27.79**) | 23 | 54.34$_{\pm 0.80}$(**-9.95**) | 60 | 41.53$_{\pm 0.80}$(**-7.27**) | 65 |
| w/o $\phi$ | 91.26$_{\pm 0.50}$(**-3.61**) | 16 | 93.85$_{\pm 0.80}$(**-2.94**) | 25 | 87.91$_{\pm 0.50}$(**-2.47**) | 15 | 61.9$_{\pm 0.60}$(**-2.39**) | 34 | 45.62$_{\pm 0.50}$(**-3.18**) | 44 |
| w/o de-MUX | 90.34$_{\pm 0.40}$(**-4.53**) | 16 | 93.07$_{\pm 0.50}$(**-3.72**) | 25 | 86.22$_{\pm 0.50}$(**-4.16**) | 15 | 61.44$_{\pm 0.50}$(**-2.85**) | 34 | 45.87$_{\pm 0.50}$(**-2.93**) | 44 |
| w/o clustering | 95.26$_{\pm 0.50}$ | 12(**-4**) | 96.80$_{\pm 0.30}$ | 20(**-5**) | 90.80$_{\pm 0.50}$ | 13(**-2**) | 64.35$_{\pm 0.80}$ | 27(**-7**) | 48.83$_{\pm 0.50}$ | 36(**-8**) |
| w/o random mask | 89.62$_{\pm 0.70}$(**-5.25**) | 16 | 94.26$_{\pm 0.70}$(**-2.53**) | 25 | 86.45$_{\pm 0.70}$(**-3.93**) | 15 | 62.90$_{\pm 0.70}$(**-1.39**) | 32 | 47.15$_{\pm 0.70}$(**-1.65**) | 44 |
| w/o PReLU | 92.43$_{\pm 0.50}$(**-2.44**) | 16 | 94.97$_{\pm 0.70}$(**-1.82**) | 25 | 87.61$_{\pm 0.50}$(**-2.77**) | 15 | 62.37$_{\pm 0.70}$(**-1.92**) | 32 | 47.23$_{\pm 0.70}$(**-1.57**) | 44 |
| w/o $\mathcal{L}_O$ | 93.15$_{\pm 0.50}$(**-1.72**) | 16 | 95.47$_{\pm 0.70}$(**-1.32**) | 25 | 89.24$_{\pm 0.50}$(**-1.14**) | 15 | 63.40$_{\pm 0.60}$ (**-0.89**) | 32 | 47.32$_{\pm 0.70}$(**-1.48**) | 44 |

aggregated output representations, (4) a variant (*w/o clustering*) that constructs the union graph by randomly grouping subgraphs without the clustering method, (5) a variant (*w/o random mask*) that removes the random mask component, and (6) a variant (*w/o PReLU*) that remains the default activation function, (7) a variant (*w/o $\mathcal{L}_O$*) that removes the regularization term $\mathcal{L}_O$. The results are presented in Table 4, and we highlight the following observations:

**Union graph retains performance stably**. As shown in Table 4, we observe that the variant *w/o union graph* exhibits a performance decrease of 25.07%, 26.45%, 27.79%, 9.95% and 7.27%, respectively, across the five datasets. The reason why the variant *w/o union graph* exhibits very poor performance is that directly superposing multiple subgraphs introduces too much interference between multiple message-passing processes. Using the union graph, MIMO-LP only superpose the message-passing operations among the overlapped regions among subgraphs, where the merged subgraphs share similar local context, effectively alleviating the interference.

**Subgraph clustering can improve the speedup ratio of MIMO-LP**. In Table 4, we observe that the variant *w/o clustering* exhibits a speedup ratio decrease of 4, 5, 2, 7 and 8, respectively, across the five datasets. Such results indicate that clustering based upon the ERR between subgraphs can deduplicate the redundant message-passing operations across subgraphs, hence effectively accelerating the running time.

**Random masking technology improves model performance in multiplexing**. In Table 4, we observe that the variant *w/o random mask* exhibits a performance decrease of 5.25%, 2.53%, 3.93%, 1.39% and 1.65%, respectively, across the five datasets. Despite the fact that random masking discards some messages, these results suggest it can alleviate message over-squashing, thereby improving model performance. We conducted a parameter analysis to explore the impact of different mask ratios on performance, with the results presented in Appendix E.3.

**The semi-linear graph encoder $\mathcal{F}_{\text{l-enc}}$ can mitigate interference in the data multiplexing**. In Table 4, we observe that the two variants (*w/o PReLU* and *w/o $\mathcal{L}_O$*) exhibit a performance decrease across the five datasets. These results indicate that the original graph encoder $\mathcal{F}_{\text{enc}}$ cannot handle multiplexing interference well.

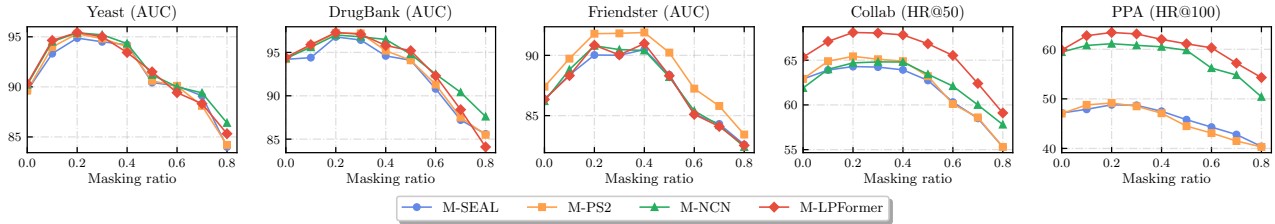

*Figure 6.* The effect of masking ratio on performance.

### E.3. Parameter Test

We conducted parameter tests on the masking ratio and subgraph pattern, with the results presented in Fig. 6 and Table 5, respectively. We highlight the following observations:

**The range of [0.2, 0.4] for the mask ratio is an optimal value interval**. In Fig. 6, we tested the impact of different masking ratios on performance across five datasets. In the range of $(0, 0.2]$, the model performance improves as the masking

*Table 5.* Performance across 1, 2, 3, and 4-hop subgraphs.

| Layers | Variants | Yeast | DrugBank | Friendster | Collab | PPA |
|---|---|---|---|---|---|---|
| l=1 | M-SEAL | $93.64_{\pm 0.50}$ | $95.13_{\pm 0.80}$ | $87.36_{\pm 0.50}$ | $63.21_{\pm 0.60}$ | $48.10_{\pm 0.50}$ |
| | M-PS2 | $93.97_{\pm 0.60}$ | $95.34_{\pm 0.80}$ | $87.50_{\pm 0.50}$ | $64.86_{\pm 0.60}$ | $48.25_{\pm 0.50}$ |
| | M-NCN | $93.45_{\pm 0.80}$ | $95.18_{\pm 1.00}$ | $88.10_{\pm 0.50}$ | $64.17_{\pm 1.00}$ | $60.92_{\pm 1.00}$ |
| | M-LPFm | $93.76_{\pm 0.80}$ | $95.92_{\pm 0.50}$ | $87.70_{\pm 0.50}$ | $67.15_{\pm 0.70}$ | $62.24_{\pm 0.60}$ |
| l=2 | M-SEAL | $94.38_{\pm 0.50}$ | $96.74_{\pm 0.80}$ | $90.10_{\pm 0.50}$ | $64.54_{\pm 0.60}$ | $48.37_{\pm 0.50}$ |
| | M-PS2 | $95.14_{\pm 0.60}$ | $97.23_{\pm 0.80}$ | $91.74_{\pm 0.50}$ | $65.29_{\pm 0.60}$ | $48.94_{\pm 0.50}$ |
| | M-NCN | $95.29_{\pm 0.80}$ | $96.94_{\pm 1.00}$ | $90.31_{\pm 0.50}$ | $64.48_{\pm 1.00}$ | $61.08_{\pm 1.00}$ |
| | M-LPFm | $95.14_{\pm 0.80}$ | $96.08_{\pm 0.50}$ | $92.59_{\pm 0.50}$ | $67.40_{\pm 0.70}$ | $62.51_{\pm 0.60}$ |
| l=3 | M-SEAL | $95.06_{\pm 0.50}$ | $96.83_{\pm 0.80}$ | $90.38_{\pm 0.50}$ | $64.56_{\pm 0.60}$ | $48.71_{\pm 0.50}$ |
| | M-PS2 | $95.43_{\pm 0.60}$ | $97.20_{\pm 0.80}$ | $91.87_{\pm 0.50}$ | $65.44_{\pm 0.60}$ | $48.83_{\pm 0.50}$ |
| | M-NCN | $95.32_{\pm 0.80}$ | $97.04_{\pm 1.00}$ | $90.42_{\pm 0.50}$ | $64.43_{\pm 1.00}$ | $61.12_{\pm 1.00}$ |
| | M-LPFm | $95.47_{\pm 0.80}$ | $96.42_{\pm 0.50}$ | $92.80_{\pm 0.50}$ | $67.46_{\pm 0.70}$ | $62.87_{\pm 0.60}$ |
| l=4 | M-SEAL | $95.04_{\pm 0.50}$ | $96.83_{\pm 0.80}$ | $90.38_{\pm 0.50}$ | $64.56_{\pm 0.60}$ | $48.70_{\pm 0.50}$ |
| | M-PS2 | $95.43_{\pm 0.60}$ | $97.20_{\pm 0.80}$ | $91.87_{\pm 0.50}$ | $65.44_{\pm 0.60}$ | $48.98_{\pm 0.50}$ |
| | M-NCN | $95.31_{\pm 0.80}$ | $97.04_{\pm 1.00}$ | $90.42_{\pm 0.50}$ | $64.43_{\pm 1.00}$ | $61.12_{\pm 1.00}$ |
| | M-LPFm | $95.43_{\pm 0.80}$ | $96.49_{\pm 0.50}$ | $92.80_{\pm 0.50}$ | $67.45_{\pm 0.70}$ | $62.87_{\pm 0.60}$ |

ratio increases. This result indicates that the masking ratio mitigates interference in data multiplexing. However, due to excessive masking severely disrupting the neighbor node information during message passing, the model performance shows a significant decline when the masking ratio $> 0.4$.

**As the hop $l$ increases, the model performance converges to the optimum**. In Table 5, we tested the performance of MIMO-LP in 1, 2, 3, and 4-hop subgraph, respectively. We observe that model performance improves as $l$ increases, which can be attributed to the richer heuristic information captured by higher-order subgraphs. However, larger subgraphs incur significantly higher computational overhead. Our empirical evaluations indicate that $l = 3$ serves as the optimal parameter, providing the best balance between performance and efficiency.

### E.4. Additional Dataset Test

In this subsection, we evaluate MIMO-LP across a broader range of datasets, where com-Orkut (Yang & Leskovec, 2012) is a social network originating from the Orkut social platform. WikiKG90Mv2 (Hu et al., 2020) is a knowledge graph originating from the Wikipedia platform. NS (Newman, 2006) is a collaboration network of researchers in network science. Random graph (Duncan J. Watts, 2012) is a synthetic random graph dataset based on the Watts-Strogatz model. 4-Regular(J, 1967) is an undirected regular graph in which every vertex has a degree of exactly four. Musae-chameleon (Yang & Leskovec, 2012) is a benchmark heterophily attributed graph, curated from a collection of blog and webpage data. Facebook (Zhang & Chen, 2018) is a social network originating from the Facebook social platform. Ecoli (Newman, 2006) is a pairwise reaction network of metabolites in Ecoli. Wikipedia (Zhang & Chen, 2018) is a knowledge graph (KG) extracted from the Wikipedia platform. Table 6 presents the statistics of these datasets.

We maintain experimental settings consistent with prior studies, with the results reported in Table 7. Notably, for WikiKG90Mv2, we adopt the benchmark's standard metric, Mean Reciprocal Rank (MRR). Specifically, MRR is defined as:

$$MRR = \frac{1}{n} \sum_{i=1}^{n} \frac{1}{rank_i}. \tag{36}$$

where $rank_i$ denotes the position of the first relevant result (ground truth) for the $i$-th query within the model's ranked list.

As observed in Table 7, there is a high ERR among subgraph samples across all datasets, suggesting that these overlaps can be eliminated to significantly accelerate both model training and inference. Variants based on MIMO-LP can eliminate redundant message propagation over overlapping topologies through superposition computation, thus enabling model acceleration. Specifically, MIMO-LP achieves a $11\times$ to $43\times$ speedup over baseline models while maintaining high accuracy across nine datasets.

*Table 6.* Statistics of the additional datasets.

| Statistics | Dataset | # of Vertices | # of Edges |
|---|---|---|---|
| Group 1 | com-Orkut | 3 072 441 | 117 185 083 |
| | WikiKG90Mv2 | 91 230 610 | 601 062 811 |
| | NS | 1589 | 2742 |
| Group 2 | Random graph | 17 572 | 35 144 |
| | 4-Regular | 5340 | 10 680 |
| | Musae-chameleon | 2277 | 31 421 |
| Group 3 | Facebook | 4039 | 88 234 |
| | Ecoli | 1805 | 14 660 |
| | Wikipedia | 4777 | 184 812 |

*Table 7.* Model comparison using SEAL, PS2, NCN and LPFormer (LPFm) as backbone model. n indicates the number of merged subgraphs. Speedup Ratio (SR) are reported against the n = 1 setting.

| Models | | | Datasets | | | | | | | | | |
| | | com-Orkut | | | | WikiKG90Mv2 | | | | NS | | |
| | n | AUC | SR↑ | ERR | n | MRR | SR↑ | ERR | n | AUC | SR↑ | ERR |
|---|---|---|---|---|---|---|---|---|---|---|---|---|
| **M-SEAL** (Ours) | 50 | $94.87_{\pm 0.50}$ | 16 | 0.78 | 100 | $11.26_{\pm 0.80}$ | 43 | 0.84 | 40 | $95.03_{\pm 0.50}$ | 15 | 0.74 |
| SEAL | 1 | $95.46_{\pm 0.20}$ | 1 | 0 | 1 | $11.84_{\pm 0.10}$ | 1 | 0 | 1 | $95.53_{\pm 0.30}$ | 1 | 0 |
| **M-PS2** (Ours) | 50 | $95.29_{\pm 0.60}$ | 16 | 0.78 | 100 | $11.24_{\pm 0.80}$ | 42 | 0.84 | 40 | $95.80_{\pm 0.50}$ | 15 | 0.74 |
| PS2 | 1 | $96.07_{\pm 0.18}$ | 1 | 0 | 1 | $11.74_{\pm 0.30}$ | 1 | 0 | 1 | $96.14_{\pm 0.30}$ | 1 | 0 |
| **M-NCN** (Ours) | 50 | $95.40_{\pm 0.80}$ | 16 | 0.78 | 100 | $10.92_{\pm 0.50}$ | 42 | 0.84 | 40 | $95.42_{\pm 0.50}$ | 15 | 0.74 |
| NCN | 1 | $96.38_{\pm 0.40}$ | 1 | 0 | 1 | $11.58_{\pm 0.40}$ | 1 | 0 | 1 | $95.49_{\pm 0.50}$ | 1 | 0 |
| **M-LPFm** (Ours) | 50 | $95.43_{\pm 0.60}$ | 15 | 0.78 | 100 | $11.29_{\pm 0.60}$ | 42 | 0.84 | 40 | $96.84_{\pm 0.90}$ | 14 | 0.74 |
| LPFm | 1 | $96.51_{\pm 0.40}$ | 1 | 0 | 1 | $11.84_{\pm 0.40}$ | 1 | 0 | 1 | $97.72_{\pm 0.30}$ | 1 | 0 |

| Models | | | Datasets | | | | | | | | | |
| | | Random graph | | | | 4-Regular | | | | Musae-chameleon | | |
| | n | AUC | SR↑ | ERR | n | AUC | SR↑ | ERR | n | AUC | SR↑ | ERR |
|---|---|---|---|---|---|---|---|---|---|---|---|---|
| **M-SEAL** (Ours) | 50 | $74.61_{\pm 0.50}$ | 11 | 0.72 | 50 | $94.87_{\pm 0.50}$ | 16 | 0.78 | 50 | $97.58_{\pm 0.80}$ | 16 | 0.79 |
| SEAL | 1 | $76.99_{\pm 0.30}$ | 1 | 0 | 1 | $95.46_{\pm 0.20}$ | 1 | 0 | 1 | $98.74_{\pm 0.10}$ | 1 | 0 |
| **M-PS2** (Ours) | 50 | $74.74_{\pm 0.50}$ | 11 | 0.72 | 50 | $95.29_{\pm 0.60}$ | 16 | 0.78 | 50 | $97.36_{\pm 0.80}$ | 16 | 0.79 |
| PS2 | 1 | $76.49_{\pm 0.30}$ | 1 | 0 | 1 | $96.07_{\pm 0.18}$ | 1 | 0 | 1 | $98.03_{\pm 0.30}$ | 1 | 0 |
| **M-NCN** (Ours) | 50 | $90.42_{\pm 0.50}$ | 11 | 0.72 | 50 | $95.40_{\pm 0.80}$ | 16 | 0.78 | 50 | $97.26_{\pm 0.50}$ | 16 | 0.79 |
| NCN | 1 | $92.47_{\pm 0.50}$ | 1 | 0 | 1 | $96.38_{\pm 0.40}$ | 1 | 0 | 1 | $98.47_{\pm 0.40}$ | 1 | 0 |
| **M-LPFm** (Ours) | 50 | $96.84_{\pm 0.90}$ | 11 | 0.72 | 50 | $95.43_{\pm 0.60}$ | 15 | 0.78 | 50 | $97.30_{\pm 0.60}$ | 16 | 0.79 |
| LPFm | 1 | $97.72_{\pm 0.30}$ | 1 | 0 | 1 | $96.51_{\pm 0.40}$ | 1 | 0 | 1 | $98.31_{\pm 0.40}$ | 1 | 0 |

| Models | | | Datasets | | | | | | | | | |
| | | Facebook | | | | Ecoli | | | | Wikipedia | | |
| | n | AUC | SR↑ | ERR | n | AUC | SR↑ | ERR | n | AUC | SR↑ | ERR |
|---|---|---|---|---|---|---|---|---|---|---|---|---|
| **M-SEAL** (Ours) | 50 | $97.25_{\pm 0.50}$ | 29 | 0.90 | 50 | $92.16_{\pm 0.50}$ | 21 | 0.85 | 50 | $88.36_{\pm 0.80}$ | 34 | 0.92 |
| SEAL | 1 | $98.13_{\pm 0.30}$ | 1 | 0 | 1 | $93.65_{\pm 0.20}$ | 1 | 0 | 1 | $89.50_{\pm 0.10}$ | 1 | 0 |
| **M-PS2** (Ours) | 50 | $97.52_{\pm 0.50}$ | 29 | 0.90 | 50 | $92.20_{\pm 0.60}$ | 21 | 0.85 | 50 | $88.19_{\pm 0.80}$ | 34 | 0.92 |
| PS2 | 1 | $98.29_{\pm 0.30}$ | 1 | 0 | 1 | $93.67_{\pm 0.18}$ | 1 | 0 | 1 | $89.04_{\pm 0.30}$ | 1 | 0 |
| **M-NCN** (Ours) | 50 | $97.10_{\pm 0.50}$ | 29 | 0.90 | 50 | $92.01_{\pm 0.80}$ | 21 | 0.85 | 50 | $88.32_{\pm 0.50}$ | 34 | 0.92 |
| NCN | 1 | $98.30_{\pm 0.50}$ | 1 | 0 | 1 | $93.54_{\pm 0.40}$ | 1 | 0 | 1 | $89.47_{\pm 0.40}$ | 1 | 0 |
| **M-LPFm** (Ours) | 50 | $96.43_{\pm 0.90}$ | 29 | 0.90 | 50 | $92.14_{\pm 0.60}$ | 21 | 0.85 | 50 | $88.41_{\pm 0.60}$ | 34 | 0.92 |
| LPFm | 1 | $97.86_{\pm 0.30}$ | 1 | 0 | 1 | $93.75_{\pm 0.40}$ | 1 | 0 | 1 | $89.43_{\pm 0.40}$ | 1 | 0 |

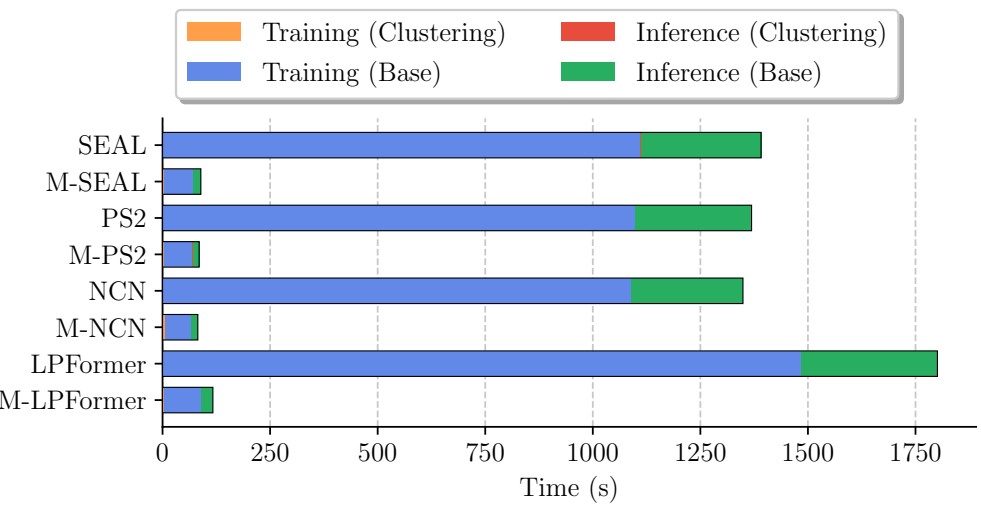

*Figure 7.* End-to-end time consumption on SEAL.

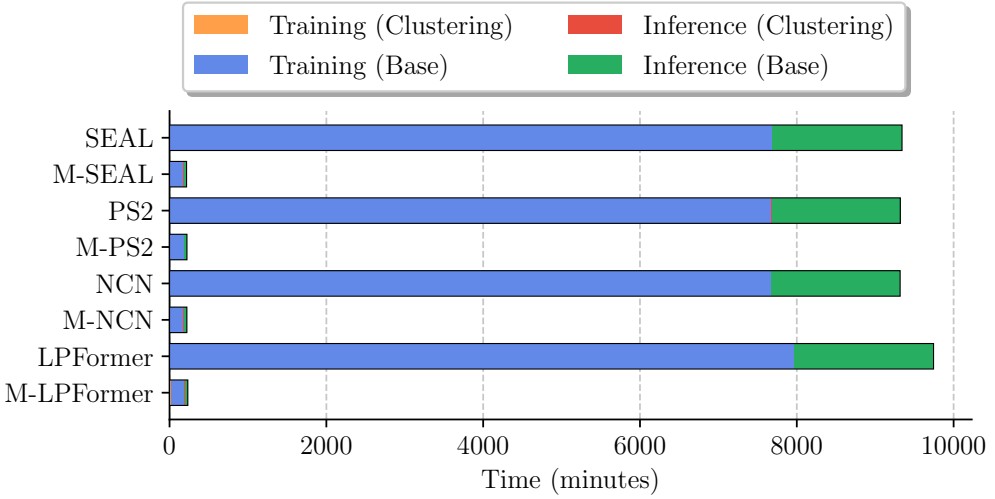

*Figure 8.* End-to-end time consumption on WikiKG90Mv2.

### E.5. Time Decomposition on Model Training and Model Inference

To observe the specific time values of different models across the four stages——clustering (training data)-training and clustering (inference data)-inference——we conducted a time decomposition experiment on the NS and WikiKG90Mv2 with the exact same experimental setup as previously used. As shown in Fig. 7 and Fig. 8, MIMO-LP accelerates both training and inference. Meanwhile, such results indicate that the clustering time is nearly negligible during both the training and inference phases of the model.

### E.6. Test on Subgraph Classification Task

Given a set of subgraphs, the subgraph classification task aims to predict the category of each subgraph by leveraging a graph encoder. To verify the generalization capability of MIMO-LP, we attempt to apply this method to the subgraph classification task. Following the study of subgraph classification (Alsentzer E, 2020), we select PPI-BP, HPO-NEURO(HPO-NR), HPO-METAB(HPO-MT) and EM-USER as benchmark datasets, with SEAL adopted as the backbone model for subgraph classification. Our experimental results are presented in Fig. 9. It is observed that MIMO-LP accelerates SEAL by a factor of $14\times$ to $22\times$ across the four subgraph classification tasks.

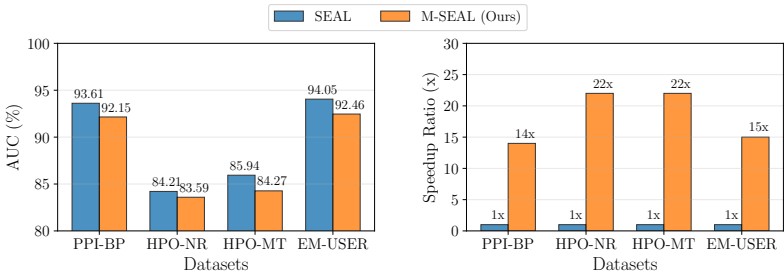

*Figure 9.* Test M-SEAL on subgraph classification task.

