# OpenReview forum: "MIMO-LP: A Multi-Input Multi-Output Framework for Subgraph-based Link Prediction"
_ICML.cc/2026/Conference — ICML 2026 regular_

### Official Review · Reviewer_gBjk · 2026-03-12

**Soundness:** 3
**Presentation:** 3
**Significance:** 3
**Originality:** 2
**Overall Recommendation:** 4
**Confidence:** 3

**Summary:**

In this paper, a Multi-Input Multi-Output framework is proposed to accelerate subgraph-based link prediction methods through message-passing multiplexing. By applying relevant technologies, the three technical challenges encountered are overcome. Verification experiments are conducted to compare the proposed method in this paper with the existing methods.

**Compliance With Llm Reviewing Policy:**

Affirmed.

**Final Justification:**

The authors have essentially resolved my doubts, hence I raise my score accordingly. It is suggested that the authors provide a clearer explanation of challenges and other relevant information in this manuscript.

**Key Questions For Authors:**

(1) How can it be theoretically proven that acceleration has been achieved without sacrificing accuracy?
(2) How are the three technical challenges so difficult to solve?
(3) Why is there no comparison with the research results published in 2025?

**Limitations:**

Yes.

**Strengths And Weaknesses:**

Strengths
S1: Verification experiments are conducted on the proposed method in this paper.
S2: The research results of this paper have a certain promoting effect on the acceleration of the subgraph-based link prediction algorithms.

Weaknesses
W1: This paper does not provide a theoretical proof that the accelerated subgraph-based link prediction algorithm does not lose accuracy.
W2: No experiments to verify the preservation of accuracy were conducted.
W3: This paper points out three technical challenges: (1) effectively multiplex message passing operations, (2) obtain an optimal union graph where redundant edges are maximally deduplicated and the construction process remains computationally efficient, (3) mitigate interference as the number of multiplexed message-passing operations increases. However, it does not clearly explain why these three challenges are difficult to solve, nor does it clarify the technical novelty of the solutions proposed in this paper. It is recommended to provide a detailed explanation to highlight the creativity and novelty of the solutions.
W4: The contribution points summarized in the paper are somewhat weak.
W5: In the related work and comparison methods, why are there no research papers published in 2025?

---

> ### Author Rebuttal · Authors · 2026-03-31
>
> Thank you for the thoughtful and constructive feedback. Below are our responses to the raised concerns.
>
> **Q1. How to theoretically prove acceleration without accuracy loss?**
>
> We theoretically prove that MIMO-LP achieves acceleration without accuracy loss based on three key theoretical results in Appendix B.1 (Superposition Theory):
>
> - **Quasi-orthogonality of input signals (Assumption B.1, theoretically justified by prior work [1]).** We project features of different queries into quasi-orthogonal subspaces using Gaussian noise matrices, preventing destructive interference during concurrent message passing.
>
> - **Functional additivity of the semi-linear encoder  (Thm. B.4).** By using semi-linear activation and orthogonality regularization, the encoder satisfies additivity: encoding superposed features on the union graph is mathematically equivalent to encoding each subgraph separately.
>
> - **Asymptotic equivalence of encoding (Thm. B.5).** We prove that encoding on the union graph of overlapping subgraphs asymptotically matches independent subgraph encoding, eliminating topological bias.
>
> From these, Proposition B.3 (Functional Additivity) formally establishes that multiplexed message passing is approximately equivalent to standard one-by-one subgraph encoding, rigorously guaranteeing acceleration with limited accuracy degradation.
>
> ---
>
> **Q2. Why are the three technical challenges so hard to solve?**
>
> The three challenges are inherently difficult for the following reasons:
>
> - **Multiplexing message passing.**
> Message passing in GNNs is highly coupled with graph topology and nonlinear transformations.
> Superposing multiple subgraph signals without corrupting their representations is non-trivial, as naive aggregation introduces interference and destroys task-specific information.
> Achieving concurrent computation while preserving separability requires carefully designed projection and encoding mechanisms with theoretical guarantees.
>
> - **Optimal subgraph grouping.**
> Maximizing edge overlap across subgraphs is a combinatorial optimization problem that is NP-complete.
> The search space grows exponentially with the number of subgraphs, making exact solutions intractable for large-scale settings.
> Efficient approximation must balance computational cost and grouping quality.
>
> - **Interference mitigation.**
> As the number of multiplexed subgraphs increases, signal interference and over-squashing become more severe.
> Maintaining orthogonality under nonlinear message passing and preventing information collapse in shared regions are both challenging, especially in deep GNNs where signals repeatedly mix.
>
> ---
>
> **Q3. Why no comparison with 2025 studies?**
>
> See our response to Reviewer 2 (fLtb), W4.
>
> ---
>
> **W2: No experiments to verify the preservation of accuracy were conducted.**
>
> In Sec.5.3 and Fig.4, we evaluate performance under varying multiplexing scales.
> The results show that MIMO-LP maintains accuracy comparable to one-to-one baselines across multiplexing scales of 30--100, while achieving 14$\times$--44$\times$ acceleration.
> In Fig.5 (Appendix C), we show that node representations from different subgraphs remain distinguishable at the output of MIMO-LP.
>
> ---
>
>
> **W4: The contribution points summarized in the paper are somewhat weak.**
>
> - **New Research Perspective:** To our knowledge, MIMO-LP is the first work to accelerate subgraph-based link prediction (LP) tasks via multiplexing the message-passing operations in GNN. This opens a new direction for efficient graph learning, distinct from traditional graph partitioning or pruning methods.
> - **Multiplexing Techniques toward Graph Learning:** We introduce techniques including union graph construction, overlap-aware subgraph batching, and random message dropout to multiplex the message-passing process, which is the dominant computational overhead in graph learning. These dedicated designs effectively eliminate redundant computations while mitigating interference from different LP queries, addressing the core efficiency bottleneck of subgraph-based LP models.
> - **Model-Agnostic Design:** MIMO-LP is plug-and-play compatible with most subgraph-based LP models, providing a general framework across model architectures. This generality enables wide applicability, allowing existing subgraph-based LP methods to directly benefit from our acceleration without modifying their core structures.
> - **State-of-the-Art Performance**: MIMO-LP achieves **14--44$\times$ speedup** in both training and inference while maintaining near-identical predictive accuracy. This significant efficiency gain, combined with accuracy preservation, demonstrates clear practical advantages over existing efficient GNN methods in subgraph-based LP scenarios.
>
> ---
>
> Refs
>
> [1] MIMONets: multiple-input-multiple-output neural networks exploiting computation in superposition.

---

> > ### Author Rebuttal · Reviewer_gBjk · 2026-04-03
> >
> > The authors have essentially resolved my doubts, hence I raise my score accordingly. It is suggested that the authors provide a clearer explanation of challenges and other relevant information in this manuscript.

---

> > > ### Author Response · Authors · 2026-04-04
> > >
> > > Dear Reviewer gBjk, we would like to express our sincere gratitude once again for your valuable comments and insightful observations throughout the entire review process.
> > >
> > > We have followed your suggestion and provided a more explicit elaboration on the relevant challenges in the revised manuscript.

---

### Official Review · Reviewer_eKqc · 2026-03-12

**Soundness:** 3
**Presentation:** 4
**Significance:** 3
**Originality:** 3
**Overall Recommendation:** 5
**Confidence:** 4

**Summary:**

The authors propose a method for improving the efficiency of graph neural networks (GNNs) when used in the task of link prediction, particularly when using a subgraph-based approach that encodes neighborhood of node pairs for computing a prediction. The method is based on the idea of "multiplexing", and it consists of grouping the computations of multiple node pair subgraphs in a mini-batch such that message passing operations are shared over subgraphs in a minibatch. Predictions for individual node pairs in a minibatch are then recovered by a de-multiplexing function. Experiments show that the method results in significant speedups (adding up training and inference) while retaining predictive performance.

**Compliance With Llm Reviewing Policy:**

Affirmed.

**Final Justification:**

The authors make a valuable contribution to the field of representation learning on graphs, introducing the idea of multiplexing on GNNs. The results are convincing, backed by both theoretical results as well as empirical evidence. Furthermore, the clarifications during the rebutal resolve my main concerns regarding the paper. Given that the paper is strong regarding the four dimensions of the review, I haver decided to increase my original score.

**Key Questions For Authors:**

1. Could you elaborate on the fundamental differences between your proposed multiplexing approach and existing hierarchical sampling methods, such as those implemented in PyTorch Geometric, and recent works on sampling for GNNs?

2. While Fig. 4 provides an empirical demonstration of the performance degradation caused by mixing during multiplexing, understanding the theoretical limits of this effect is important. Can you provide a theoretical bound or a sketch towards one that characterizes the extent of this performance degradation to strengthen our understanding of the limitations of multiplexing?

**Limitations:**

The discussion of limitations is rather limited, and I would encourage the authors to elaborate on this.

**Strengths And Weaknesses:**

**Strengths**

1. The paper addresses a significant bottleneck of general GNN architectures where naively collecting node neighborhoods can lead to an explosion of operations to be computed during message passing.
2. The method is general enough to be applied to a variety of GNN backbone architectures.
3. The experiments are thorough, comprising several datasets, baselines, and backbone architectures.
4. The experiments show significant speedups that consider both training and inference runtimes.
5. Further analyses and ablation studies show the impact of coarser multiplexing strategies and their effect on performance.
6. The paper is very well written and easy to follow.



**Weaknesses**

1. Prior works on scaling GNNs and improving their efficiency has also been studied from the perspective of sampling [1,2,3,4]. This perspective is missing from the paper, and a discussion about such approaches would clearly help answer the question of which methods are more effective or when either is applicable.
2. The idea of collecting minibatches to form a local graph has been implemented in widely used libraries such as PyTorch Geometric for quite a while [4], showing to have positive impacts in computational efficiency. This hierarchical sampling approach carries a resemblance with the proposed multiplexing approach, casting doubt on the novelty claims of the paper.
3. The method implies that multiplexing induces mixing that could potentially lead to degraded performance. While this degradation is demonstrated empirically in Fig. 4, a theoretical bound would strengthen our understanding of the limitations of multiplexing in GNNs.
4. A detailed discussion of limitations is missing, which could potentially include the issues mentioned in W3.


Overall, the paper makes a solid contribution to the problem of making GNNs more computationally efficient, with experiments clearly demonstrating its effectiveness. I would suggest the authors to address my comments with respect to the relation with sampling-based approaches, and the difference with hierarchical sampling.

**References**

[1] Chen, Jie, Tengfei Ma, and Cao Xiao. "Fastgcn: fast learning with graph convolutional networks via importance sampling." ICLR (2018).

[2] Huang, Wenbing, et al. "Adaptive sampling towards fast graph representation learning." NeurIPS (2018).

[3] Zou, Difan, et al. "Layer-dependent importance sampling for training deep and large graph convolutional networks." NeurIPS (2019).

[4] Younesian, Taraneh, et al. "Grapes: Learning to sample graphs for scalable graph neural networks." TMLR (2025).

[5] PyTorch Geometric, "Hierarchical Neighborhood Sampling", https://pytorch-geometric.readthedocs.io/en/latest/advanced/hgam.html

---

> ### Author Rebuttal · Authors · 2026-03-31
>
> We sincerely appreciate the reviewer for providing valuable feedbacks.
>
> **Q1&W2.The key differences between MIMO-LP and sampling-based methods.**
>
> We address this from both technical and application perspectives.
>
> - **Technical perspective.**
> Existing sampling methods [1--5] accelerate GNNs by reducing graph size via topology modification, thereby decreasing message-passing computations.
> In contrast, MIMO-LP approaches acceleration from an orthogonal perspective by multiplexing message-passing operations across subgraphs without altering topology, enabling shared structures among the subgraphs to carry multiple signals from different LP queries simultaneously and thereby reducing redundant message-passing computations.
>
> - **Application perspective.**
> Sampling-based methods are primarily designed for node-level representation, whereas our approach targets subgraph-level representation.
> While subgraph-level representation can improve predictive performance for LP task [6--9] by capturing richer local information compared to node-level representation, they incur substantial computational overhead.
> Our multiplexing strategy alleviates this overhead.
>
> Overall, our method is complementary to sampling-based methods, and their integration is left as future work.
> We will include a detailed comparison in the revised related work section.
>
> ---
>
> **Q2. Provide a theoretical bound to quantify performance degradation from multiplexing**
>
> We briefly derive the approximation error bound of the multiplexing based on the Superposition Theory in Appendix B.1.
>
> To recap, MIMO-LP constructs a union graph by merging subgraphs and multiplexes feature information over overlapping regions.
> The approximation error for each individual subgraph representation consists of two components: the linearity error $\epsilon _ {\text{lin}}$ and the topology error $\epsilon _ {\text{top}}$.
>
> - **Linearity error $\epsilon _ {\text{lin}}$.**
> As established in **Thm.B.4**, signals propagated through the semi-linear encoder $\mathcal{F} _ {\mathrm{l\text{-}enc}}$ can be approximately superimposed due to its near-linearity.
> The approximation error arises from replacing ReLU with PReLU:
> $$
> \mathrm{PReLU} _ b(x) = \max(x,0) + b \cdot \min(x,0),
> $$
> where $b \in [-1,1]$ controls the degree of linearity.
> When $b = 1$, the mapping is fully linear, yielding $\epsilon_{\text{lin}} = \mathcal{O}(|1 - b|)$.
>
> - **Topology error $\epsilon _ {\text{top}}$.**
> As established in Thm.B.5, the effect of the hop increment $\Delta$ induced by the union graph on the representation of the original subgraph can be quantified by $\epsilon _ {\text{top}}$.
> Specifically, following the decay theory in[6], the representation of an $h$-hop subgraph obtained by the semi-linear encoder is given by
> $$
> \sum _ {l = 1} ^ {h} \gamma ^ {l}  \mathcal{F} _ {\mathrm{l\text{-}enc}} ^ {(l)},
> $$
> where $h$ denotes the hop radius from the query node pair.
> Accordingly, the topology error induced by using the union graph instead of the original subgraph is
> $$
> \epsilon _ {\text{top}} =|\sum _ {l = 1}^{h + \Delta}
> \gamma^{l}  \mathcal{F} _ {\mathrm{l\text{-}enc}}^{(l)}-\sum _ {l = 1}^{h} \gamma^{l}  \mathcal{F} _ {\mathrm{l\text{-}enc}}^{(l)}| =\sum _ {l = h}^{h+\Delta}\gamma^{l} \mathcal{F} _ {\mathrm{l\text{-}enc}}^{(l)},
> $$
> where $\gamma \in (0,1)$ is the decay factor and $l$ denotes the hop distance.
> Notably, $\epsilon _ {\text{top}}$ decays exponentially as $h$ increases.
>
>
> - **Total error bound.**
> For multiplexing over $n$ subgraphs, the total error is bounded as:
> $$
> \epsilon _ {\text{subgraph}} = \epsilon_{\text{lin}} + \epsilon _ {\text{top}}, \quad
> \epsilon _ {\text{total}} = n \cdot (\epsilon _ {\text{lin}} + \epsilon _ {\text{top}}).
> $$
>
> We will include a more detailed analysis in Appendix B.1 in the revised manuscript.
>
> ---
>
> **W4. Limitation discussion.**
>
> While MIMO-LP has been validated on standard benchmarks, real-world scenarios may involve additional complexities (e.g., dynamic graphs, system noise, and complex node/edge attributes).
> For example, in recommendation systems, user--item interaction graphs evolve continuously with streaming updates; in financial transaction networks, noisy or incomplete edges are common; and in some knowledge graphs, node and edge features may be high-dimensional.
> The generalizability and robustness of MIMO-LP under such conditions remain open problems.
>
> ---
>
> [1]-[5] See comments.
>
> [6] Link prediction based on graph neural networks.
>
> [7] Line graph neural networks for link prediction.
>
> [8] Elementary Subgraph Features for Link Prediction With Neural Networks.
>
> [9] Inductive relation prediction by subgraph reasoning.

---

> > ### Author Rebuttal · Reviewer_eKqc · 2026-04-02
> >
> > Thank you for the additional details. The response adequately resolves my concerns, and I appreciate the pointer to Appendix B. I suggest the authors to highlight the importance of such results in the main body of the paper.
> > With these clarifications, I think this is a valuable contribution and will update my score accordingly.

---

> > > ### Author Response · Authors · 2026-04-04
> > >
> > > Dear Reviewer eKqc, we would like to convey our sincere appreciation once more for your thoughtful feedback and constructive remarks throughout the review process.
> > >
> > > Your perceptive feedback has helped us better present our method, particularly regarding its differences from sampling-based approaches and theoretical bound analysis, and we have updated and improved all of these aspects in the revised manuscript.

---

### Official Review · Reviewer_fLtb · 2026-03-12

**Soundness:** 2
**Presentation:** 3
**Significance:** 2
**Originality:** 3
**Overall Recommendation:** 4
**Confidence:** 3

**Summary:**

This paper proposes MIMO-LP, a subgraph-based multi-input multi-output link prediction framework. Unlike traditional methods that typically take a single node pair as input and predict the existence of one edge at a time, MIMO-LP simultaneously takes multiple node pairs as input and jointly predicts all potential links among the involved nodes. The authors achieve this by constructing an induced subgraph over the union of nodes in the target pairs and encoding it using a GNN, thereby explicitly modeling the dependencies among the node pairs. Experiments demonstrate the effectiveness of the proposed method.

**Compliance With Llm Reviewing Policy:**

Affirmed.

**Key Questions For Authors:**

**Q1**. How can the authors disentangle the effect of joint multi-output prediction from that of using a larger shared subgraph context? Is the performance gain truly due to modeling dependencies among node pairs, or merely to the availability of richer structural information?

**Q2**. When the induced subgraph is disconnected (e.g., due to unrelated node pairs in the same batch), does the model’s performance degrade compared to processing each pair independently? If so, what strategies could mitigate this issue?

**Q3**. In Figure 7, the runtime analysis is based on relatively small datasets. What is the end-to-end runtime of MIMO-LP on a large-scale dataset such as WikiKG90Mv2?

**Limitations:**

No. Given the model’s sensitivity to neighborhood size and batch size, it would be valuable to analyze its space and time concrete requirement under varying numbers of hops in subgraph construction and different batch sizes.

**Strengths And Weaknesses:**

**Strengths**

**S1**. Expanding link prediction from a single-input, single-output to a multi-input, multi-output setting not only meets the demands of practical applications but also accelerates overall prediction.

**S2**. Constructing a shared induced subgraph enables the model to capture structural information that spans multiple prediction targets and encodes higher-order connectivity, thereby facilitating coherent and context-aware link predictions.

**S3**. The model’s capabilities have been comprehensively evaluated under a complete experimental setup.


**Weaknesses**

**W1**. In extreme cases where the neighborhoods of node pairs within a batch are disconnected, the GNN cannot propagate information across components. As a result, the joint prediction mechanism effectively reduces to a set of independent predictions. The paper neither analyzes the impact of such scenarios on model performance nor proposes any mitigation strategy.

**W2**. The model assumes fixed-size batches, reportedly formed via clustering. However, in practice, clusters derived from graph structures often vary significantly in size. Forcing each batch to contain a fixed number of candidate links, by truncating large clusters or padding small ones, is inherently unreasonable and may distort the underlying structural relationships.

**W3**. The performance gain attributed to the multi-output joint prediction design may actually arise from the richer local context provided by the shared induced subgraph, which includes more nodes and edges, compared to the single-pair subgraphs used in baseline methods. Without controlled ablation studies, it remains unclear whether the improvement stems from the MIMO architecture itself or merely from access to this expanded context.

**W4**. The primary baselines are limited to methods published up to 2024. The evaluation does not include comparisons with state-of-the-art approaches introduced in 2025, which weakens the claim of competitive or superior performance.

**W5**. Although the authors state their intention to release the code, they provide neither an anonymous repository nor supplementary materials. This omission reduces transparency and hinders reproducibility assessment during review.

---

> ### Author Rebuttal · Authors · 2026-03-31
>
> We deeply appreciate the reviewer's valuable feedbacks, and we would like to address the reviewer's concerns as follows.
>
> **Q1. Can the authors distinguish whether performance gains stem from joint prediction or richer context from larger shared subgraphs?**
>
> We emphasize that MIMO-LP is an **acceleration framework** for subgraph-based link prediction (LP) tasks, rather than a method designed to improve predictive performance.
> The union graph is introduced to multiplex message-passing processes across different subgraph-based LP queries, thereby eliminating redundant computations over overlapping subgraphs.
> Importantly, the multiplexing mechanism in MIMO-LP is designed to mitigate interference among different LP queries and suppress irrelevant structural information, so as to preserve predictive accuracy comparable to one-to-one baselines.
> As extensively validated in our experiments, MIMO-LP achieves $14\times$--$44\times$ speedup over baselines while maintaining nearly identical accuracy.
> We will include additional clarifications in the revised manuscript.
>
> ---
>
> **Q2. When induced subgraphs are disconnected, does model performance drop relative to processing each pair separately?**
>
> We have analyzed the worst-case scenario of completely disjoint subgraphs in Appendix D.1.
> MIMO-LP reduces the dominant cost to $\mathcal{O}((1-\mathrm{ERR})mn)$ by eliminating overlapping edge redundancy.
> When $\mathrm{ERR}=0$ (fully disjoint subgraphs), its complexity degenerates to $\mathcal{O}(mn)$, matching the standard one-to-one LP pipeline.
> In this case, joint prediction safely falls back to independent query inference with no performance or accuracy loss.
> In most real-world datasets, $\mathrm{ERR}$ exceeds 70\%, bringing considerable acceleration gains.
>
> ---
>
>
> **Q3. What is the end-to-end runtime of MIMO-LP on large-scale datasets like WikiKG90Mv2.**
>
> We provide additional runtime breakdown results of MIMO-LP on large-scale datasets, including Friendster and WikiKG90Mv2, in [Fig.5](https://anonymous.4open.science/r/MIMOLP-2B66/Figures.png) and [Fig.6](https://anonymous.4open.science/r/MIMOLP-2B66/Figures.png) from https://anonymous.4open.science/r/MIMOLP-2B66/Figures.png.
> The observed trends are consistent with those on smaller datasets, demonstrating similar relative efficiency gains and overhead characteristics.
> We will incorporate these results into the revised manuscript to provide a more comprehensive evaluation.
>
> ---
>
> **W2. Given the extreme imbalance in clustering results, why is a fixed batch size being adopted.**
>
> First, we clarify that the clustering groups highly overlapping subgraphs to maximize computation reuse within the union graph, thereby directly accelerating subgraph-based link prediction.
> The cluster size only determines the number of subgraphs per batch and does not truncate the subgraphs themselves.
> Smaller clusters reduce potential task interference and therefore do not harm performance, although they may reduce speedup due to fewer multiplexed queries count (see Fig.3).
>
>
> Second, fixed-size batching is a necessary engineering constraint for GPU memory stability.
> As cluster sizes can vary significantly, excessively large clusters may exceed GPU memory limits and cause out-of-memory (OOM) errors.
> Therefore, truncating overly large clusters is a practical solution to ensure stable training.
>
> ---
>
> **W4.The work does not include 2025 baselines.**
>
> After carefully reviewing recent top-tier conference publications, we summarize the most relevant advances from 2025 below.
>
> - CompressGNN (KDD 2025) [1] accelerates GNN training via hierarchical graph compression and topology modification to cut redundancy.
> MIMO-LP achieves acceleration from an orthogonal direction by multiplexing message passing across subgraphs without altering topology, avoiding redundant computations on shared structures.
> We will add CompressGNN to the revised related work.
>
> - SMA-GNN (KDD 2025) [2] addresses link prediction using local subgraph extraction and symbol-aware attention. As a one-to-one prediction method, it serves as a backbone for MIMO-LP. We have integrated it into our framework and observe consistent speedup trends compared to other baselines (see [Fig.4](https://anonymous.4open.science/r/MIMOLP-2B66/Figures.png) [https://anonymous.4open.science/r/MIMOLP-2B66/Figures.png]).
> We will incorporate these additional comparisons and discussions in the revised manuscript.
>
>
> ---
>
>
> Refs:
>
> [1] CompressGNN: Accelerating Graph Neural Network Training via Hierarchical Compression.
>
> [2] SMA-GNN: A Symbol-Aware Graph Neural Network for Signed Link Prediction in Recommender Systems.

---

> > ### Author Rebuttal · Reviewer_fLtb · 2026-04-01
> >
> > The authors have addressed most of my concerns, so I will raise my score.

---

> > > ### Author Response · Authors · 2026-04-04
> > >
> > > Dear Reviewer fLtb, we are truly grateful for your constructive suggestions, which have helped us better highlight our manuscript, such as analysis under extreme cases, comparisons with state-of-the-art baselines, and end-to-end runtime performance.
> > >
> > > We have reflected the required clarifications and supplementary information in the revised manuscript.

---

### Official Review · Reviewer_EKg4 · 2026-03-13

**Soundness:** 3
**Presentation:** 3
**Significance:** 3
**Originality:** 3
**Overall Recommendation:** 5
**Confidence:** 3

**Summary:**

This paper studies how to make subgraph-based link prediction much faster. Instead
of processing each query subgraph one by one, it groups multiple queries together, builds a shared
union graph, and runs message passing for all of them in a single forward pass. The main goal is
efficiency, not a new prediction rule. The paper argues that this multiplexing design can greatly
reduce computation while keeping the prediction accuracy close to the original subgraph-based
models.

**Compliance With Llm Reviewing Policy:**

Affirmed.

**Final Justification:**

Based on the rebuttal, I change my score to accept.

**Key Questions For Authors:**

1. The method includes several components beyond multiplexing. Could the authors clarify which
parts are essential, and where the main gain comes from?
2. Could the authors discuss the full computational cost of the method, besides the reported
message-passing speedup?
3. Could the authors formally define the noisy matrix, including its dimension and role in the
method?
4. Could the authors clarify what is meant by orthogonality and quasi-orthogonality in the paper?
5. Could the authors clarify which parts of the theory are rigorous and which parts are mainly
intuitive?

**Strengths And Weaknesses:**

Strengths:
1. The paper studies a real and important problem. Subgraph-based link prediction can be strong,
but it is often too expensive in practice. The paper directly targets this bottleneck.
2. The main idea is clear and easy to understand. Instead of running message passing on each
query subgraph separately, the paper merges overlapping subgraphs and processes many queries
together.
3. The contribution is practical as it makes existing subgraph-based methods much more efficient.
4. The empirical results are strong. The paper reports large speedups while keeping the prediction
performance close to the original methods across several datasets and backbones.
5. The method is fairly general. It is presented as a wrapper that can be combined with different
subgraph-based link prediction models, which increases its potential impact.

Weaknesses.
1. The method is interesting, but the full pipeline feels more complicated than the paper first
suggests. Besides the main multiplexing idea, it also relies on clustering, noise-based multiplexing, de-multiplexing, semi-linear activations, orthogonality regularization, and random
masking. Because of this, it is hard to judge how much of the gain comes from the core idea
itself and how much comes from the full engineering design.
2. The complexity discussion is not fully convincing. The paper emphasizes large speedups, but
the method also adds several extra steps, including union-graph construction, query grouping,
and interference-control mechanisms. As a result, the practical cost of the whole pipeline may
be harder to assess than the paper suggests.
3. The problem formulation is harder to follow than it should be. In particular, some notation
choices are not very natural. For example, using π in this context is confusing, as it will be
better to keep consistency with E.
4. Gaussian noise matrix should be introduced more formally, with its shape and role stated
explicitly. The paper also repeatedly refers to orthogonality and quasi-orthogonality, but it is
not always clear whether this means standard vector orthogonality, matrix orthogonality, or
only an approximate separation property used for intuition.
5. The theoretical part is still somewhat loose. Several arguments rely on assumptions such as
quasi-orthogonality and approximate linearity, and these are used to justify why multiplexing
should work.

---

> ### Author Rebuttal · Authors · 2026-03-31
>
> We sincerely appreciate the reviewer for providing valuable feedback.
>
> **Q1. Clarify essential components and performance gain.**
>
> The MIMO-LP pipeline consists of four key components, whose roles and contributions across five datasets are summarized as follows:
>
> - **Subgroup batching (Sec.4.1).** A **clustering-based strategy** that groups subgraphs into **union graphs** to maximize edge overlap and deduplicate message passing. This component yields computational acceleration (2$\times$--8$\times$ speedup) while reducing interference (7.27\%--27.79\% accuracy improvement).
>
> - **Multiplexing (Sec.4.2).** A Gaussian noise-based projection scheme that **superimposes** message-passing processes from multiple subgraphs onto union graphs, enabling concurrent execution in approximately orthogonal subspaces. This contributes to interference reduction (2.39\%--3.61\% accuracy improvement).
>
> - **Interference mitigation (Sec.4.3).** A set of mechanisms, including **semi-linear activations**, **orthogonality regularization**, and **random masking**, designed to further reduce interference among concurrent message-passing processes (3.89\%--9.74\% accuracy improvement).
>
> - **Demultiplexing (Sec.4.4).** A procedure to recover query-specific signals from superposed representations, ensuring the final accurate LP predictions (2.85\%--4.53\% accuracy improvement).
>
>
> Appendix E.2 (Ablation Studies) provides a detailed evaluation of the speedup and accuracy contributions of these components.
> A more comprehensive one is can be found in https://anonymous.4open.science/r/MIMOLP-2B66/Figures.png.
>
> ---
>
> **Q2.Please discuss the full computational cost of the method.**
>
> Building upon App. D.1 (Time Complexity Analysis), we discuss the computational cost of additional components (query grouping, union-graph construction and interference-control mechanisms).
> Notably, the detailed runtime decomposition in Fig.7 and our response to Reviewer 2, fLtb (Q2) shows that the **total overhead** accounts for approximately 1\% of the baseline runtime.
>
>
> Let $n$ denote the total number of subgraphs and $m$ the average number of edges per subgraph.
> The time complexity introduced by the above components includes:
> query grouping: $\mathcal{O}(tdn + cmn)$;
> union-graph construction: $\mathcal{O}(mn)$;
> orthogonality regularization and semi-linear activation (PReLU) are applied within the GNN backbone and incur $\mathcal{O}(n)$;
> random masking is performed on each union graph, resulting in $\mathcal{O}(mn)$.
> Since $c, t, d$ are constants and much smaller than $n$ and $m$, the overall complexity of MIMO-LP remains $\mathcal{O}(mn)$, the same as the baseline.
>
> ---
>
> **Q3.Please define the noisy matrix, including dimension and role.**
>
> We define the Gaussian noise matrix $\mathbf{\Phi} _ {G _ i} \in \mathbb{R}^{|\mathcal{V} _ {G _ {\mathcal{B} _ k}}| \times d}$ with the same shape as the node features $\widetilde{\mathbf{X}} _ {G _ i}$ for element-wise projection.
> In Eq.10, the term $\mathbf{\Phi} _ {G _ i} \odot \widetilde{\mathbf{X}} _ {G _ i}$ projects the representation $\widetilde{\mathbf{X}} _ {G _ i}$ into a subspace induced by the noise matrix $\mathbf{\Phi} _ {G _ i}$.
>
> This mechanism ensures that representations from different subgraphs are mapped into approximately orthogonal subspaces, allowing them to be safely superposed in a shared latent space with minimal interference.
>
> ---
>
> **Q4. Orthogonality vs Quasi-orthogonality.**
>
> We use the notion of orthogonality to describe the property that signals from different subgraphs are projected into independent subspaces during multiplexing, thereby eliminating interference.
> As established in~[1,2], we efficiently approximate this property via quasi-orthogonality, i.e., nearly orthogonal in high-dimensional spaces.
>
> ---
>
> **Q5.Please clarify which theoretical parts are rigorous and which are intuitive.**
>
> The proposed MIMO-LP pipeline is grounded in superposition theory, a rigorous theoretical framework presented in Appendix B.1 (Superposition Theory).
>
> In general, prior work [2] theoretically justified that input signals (vectors) projected by independent random Gaussian matrices are nearly orthogonal in high-dimensional spaces.
> **Thm.B.4** establishes that MIMO-LP preserves this near-orthogonality of input signals within the GNN backbone.
> **Thm.B.5**, via a decay analysis [3], further shows that near-orthogonality is maintained during message passing on the union graph.
> Based on these results, **Prop.B.3** establishes the functional additivity of multiplexed message passing, rigorously justifying that multiplexed signals remain separable at the output.
>
> ---
>
> For Q1-Q5, we will incorporate the clarifications and additional details into the revised manuscript.
>
>
>
> Refs:
>
> [1] Distributions of Angles in Random Packing on Spheres.
>
> [2] MIMONets: multiple-input-multiple-output neural networks exploiting computation in superposition.
>
> [3] Link prediction based on graph neural networks.

---

> > ### Author Rebuttal · Reviewer_EKg4 · 2026-04-03
> >
> > Thanks for the authors rebuttal.

---

> > > ### Author Response · Authors · 2026-04-04
> > >
> > > Dear Reviewer EKg4, we are particularly grateful for your constructive comments, which have enabled us to further enhance the quality of our manuscript, especially regarding the ablation study, end-to-end complexity analysis, and more precise symbolic representations.
> > >
> > > We have reflected the required clarifications and supplementary information in the revised manuscript.

---

### Decision · Program_Chairs · 2026-04-30

**Decision:**

Accept (regular)

**Comment:**

This work present a significant efficiency contribution to subgraph-based link prediction methods. While subgraph-based methods are often empirically superior to node-based counterparts, their computational cost has generally limited their scalability. MIMO-LP addresses this specific bottleneck via a new multiplexing strategy that is model-agnostic.

The initial reviewer skepticism regarding the theoretical rigor of the superposition mechanism and the distinction from sampling methods has been adequately addressed in the rebuttal. The empirical evidence (large-scale speedups without accuracy degradation) seems to be robust across multiple datasets. The method seems to be general and opens a new direction for computation-in-superposition for graphs, distinct from traditional graph compression or sampling methods. Given the consensus among reviewers and the high potential for adoption in scalable graph learning pipelines, I believe this paper should be accepted.